# Genetic determinants of cellular addiction to DNA polymerase theta

Wanjuan Feng[1], Dennis A. Simpson[1], Juan Carvajal-Garcia [1], Brandon A. Price[1], Rashmi J. Kumar[1], Lisle E. Mose[1], Richard D. Wood [2], Naim Rashid[1,3], Jeremy E. Purvis[4], Joel S. Parker [1,4], Dale A. Ramsden[1,5] & Gaorav P. Gupta [1,5,6]

Polymerase theta (Pol θ, gene name *Polq*) is a widely conserved DNA polymerase that mediates a microhomology-mediated, error-prone, double strand break (DSB) repair pathway, referred to as Theta Mediated End Joining (TMEJ). Cells with homologous recombination deficiency are reliant on TMEJ for DSB repair. It is unknown whether deficiencies in other components of the DNA damage response (DDR) also result in Pol θ addiction. Here we use a CRISPR genetic screen to uncover 140 *Polq* synthetic lethal (PolqSL) genes, the majority of which were previously unknown. Functional analyses indicate that Pol θ/TMEJ addiction is associated with increased levels of replication-associated DSBs, regardless of the initial source of damage. We further demonstrate that approximately 30% of TCGA breast cancers have genetic alterations in PolqSL genes and exhibit genomic scars of Pol θ/TMEJ hyperactivity, thereby substantially expanding the subset of human cancers for which Pol θ inhibition represents a promising therapeutic strategy.

[1] Lineberger Comprehensive Cancer Center, University of North Carolina at Chapel Hill, Chapel Hill, NC 27599, USA. [2] Department of Epigenetics and Molecular Carcinogenesis, University of Texas MD Anderson Cancer Center, Smithville, TX 78957, USA. [3] Department of Biostatistics, University of North Carolina at Chapel Hill, Chapel Hill, NC 27599, USA. [4] Department of Genetics, University of North Carolina at Chapel Hill, Chapel Hill, NC 27599, USA. [5] Department of Biochemistry and Biophysics, University of North Carolina at Chapel Hill, Chapel Hill, NC 27599, USA. [6] Department of Radiation Oncology, University of North Carolina at Chapel Hill, Chapel Hill, NC 27599, USA. Correspondence and requests for materials should be addressed to G.P.G. (email: gaorav@med.unc.edu)

DNA double strand breaks (DSBs) arise spontaneously during DNA replication or upon exposure to exogenous clastogens and threaten both genome integrity and cellular viability[1–3]. Efficient and accurate DSB repair is thus vital for cancer prevention and organismal survival. DSB repair pathways are broadly classified into two categories: homology-directed repair (HDR) and non-homologous end joining (NHEJ). HDR requires 5′ to 3′ end resection, Rad51 loading, strand invasion, and DNA synthesis using an intact homologous template[4]. In contrast, classical NHEJ (c-NHEJ) does not require a homologous template and is dependent on the Ku complex, DNA-PK, and XRCC4/Ligase 4[5]. An alternative end joining (alt-EJ) pathway has also been described, but unlike c-NHEJ, alt-EJ acts on the same 5′ to 3′ resected DSBs that are intermediates in HR. Alt-EJ employs a synthesis-dependent mechanism that is directed by short tracts of flanking microhomology (MH)[1,6], giving rise to a characteristic pattern of MH-flanked deletions and/or templated insertions. Several genes have been implicated in alt-EJ, including 5′ to 3′ resection factors (e.g., Mre11, Rad50, Nbn, CtiP, and Exo1), PARP1, and LIG3. However, the gene that is most specifically linked to Alt-EJ is the A-family DNA Polymerase θ (Pol θ, gene name *Polq*)[1,7]. Alt-EJ signatures at chromosomal breaks are substantially reduced in *Polq*[−/−] cells from diverse metazoan and plant organisms[8–10]. Thus, Pol θ has emerged as the predominant mediator of alt-EJ, and this alternative DSB repair pathway has been designated Theta Mediated End Joining (TMEJ)[9,11].

TMEJ is intrinsically an error-prone pathway, yet its evolutionary conservation in metazoans and plants suggests that it likely has a physiological role in promoting genome integrity[1]. Indeed, *Polq*[−/−] cells demonstrate elevated levels of spontaneous DNA damage[12]. A prior study suggested that TMEJ competes with HDR for DSB repair[13], but this model does not explain how TMEJ may promote genome stability. In *C. elegans*, TMEJ has an important role in the repair of replication-associated DSBs, particularly at G-quadruplex (G4) structures[14,15]. In that study, Pol θ deficiency resulted in large-scale deletions at chromosomal G4 sites. However, the physiological role of TMEJ in promoting genome integrity in mammals remains unclear.

In normal cells, TMEJ accounts for a small minority of DSB repair[10]. Consistent with a limited role in global DSB repair, *Polq* deficiency has a relatively minor impact on organismal development in flies[16], worms[17], and mice[12]. However, recent studies have demonstrated that *Polq* nevertheless becomes essential in cells with deficiency in canonical DSB repair pathway genes (*Brca1*, *Brca2*, and *Ku70*), indicating synthetic lethal genetic interactions that are consistent with an essential role for Polθ/TMEJ as a backup to repair by either HR or NHEJ[10,13,18]. This observation has resulted in enthusiasm for Pol θ as a therapeutic target in breast and ovarian cancers with *BRCA1/2* deficiency[19]. However, it remains unknown whether *Polq* is also synthetic lethal with other genes in the HR and NHEJ pathways, and more broadly, with other genetic mediators of the DNA damage response (DDR) pathway. Here, we report findings from a synthetic lethal CRISPR screen to identify DDR gene mutations that induce cellular addiction to Pol θ. We uncover a broad landscape of synthetic lethality with *Polq*, and provide evidence that this reflects a critical role for Pol θ in protecting cells from accumulation of non-productive HR intermediates at sites of DNA replication-associated DSBs. Finally, we find that human breast cancers with mutations in *Polq* synthetic lethal (PolqSL) genes identified in our CRISPR screen may be addicted to Pol θ, based on increased expression of TMEJ-associated genomic scars.

## Results

**CRISPR synthetic lethal screens.** To gain broader insight into the contexts where Pol θ-mediated genome maintenance is essential for cellular viability, we performed a CRISPR loss of function screen in *WT*, *Polq*[−/−], and *Polq*[hPOLQ] (*Polq*[−/−] reconstituted with human *POLQ*) MEF cell lines, which were described previously and functionally validated[10,12,20]. *Polq*[−/−] MEFs have a normal cell cycle profile[10], yet exhibit elevated levels of spontaneous chromosomal aberrations that are reversed after complementation with human *POLQ* (Supplementary Fig. 1). The goal of the CRISPR screen was to identify gene mutations that are tolerated in *WT* and *Polq*[hPOLQ] MEFs yet lethal in *Polq*[−/−] MEFs, thereby indicative of a synthetic lethal genetic interaction. A custom synthesized "DDR-CRISPR" lentiviral library was used for the screen, which targets 309 murine DDR genes with 10 small guide RNAs (sgRNAs) per gene and also includes 834 non-targeting sgRNA controls (Supplementary Data 1). For each biological replicate, $2 \times 10^6$ MEFs were transduced with the DDR-CRISPR lentiviral library at low multiplicity of infection (<1), and passaged for 8 population doublings prior to genomic DNA isolation (Fig. 1a). High-throughput sequencing (average 250× read depth) was used to quantify the abundance of each sgRNA sequence relative to all mapped reads, similar to previously described methods[21] (Fig. 1a). A "Gene Abundance Change Score" was calculated as described in the methods. Thresholds for statistical significance were established by using the set of control sgRNAs as an internal control for abundance changes that are due to off-target effects (see "Methods").

Plotting the Gene Abundance Change Scores, we observed a striking depletion of many DDR gene-targeting sgRNAs in *Polq*[−/−] relative to *WT* MEFs (Fig. 1b). In contrast, control sgRNAs were not depleted in *Polq*[−/−] cells and, in fact, were enriched relative to their abundance in *WT* cells due to the depletion of a large number of DDR gene-targeting sgRNAs. Moreover, the vast majority of sgRNA abundance changes in *Polq*[−/−] MEFs could be definitively attributed to Pol θ deficiency, as they were not observed when *Polq*[−/−] cells were reconstituted with WT human *POLQ* (*Polq*[hPOLQ]) (Fig. 1b). To mitigate any clone-specific genetic interactions we directly compared Gene Abundance Change Scores in *Polq*[−/−] MEFs relative to *Polq*[hPOLQ] MEFs, and identified 142 significant genetic interactions using two complementary statistical tests (Fig. 1c). All but two of these genes (140 total) had corresponding sgRNAs that were depleted in *Polq*[−/−] MEFs relative to reconstituted *Polq*[hPOLQ] MEFs, and thus classified as *Polq* synthetic lethal (PolqSL) genes.

Due to the large proportion of *Polq* synthetic lethal gene interactions identified in our screen (45% of 309 genes evaluated), we performed two additional control experiments. First, we conducted the same DDR-CRISPR screen in an immortalized MEF line that is deficient in another DNA repair polymerase, Pol μ (*Polm*), that participates in NHEJ repair[22]. We did not identify any statistically significant synthetic lethal gene interactions with *Polm* deficiency (Supplementary Fig. 2a, b and Supplementary Data 3), indicating the broad landscape of DDR gene synthetic lethality is not observed for all DNA repair-associated polymerases. To address whether *Polq*[−/−] cells are prone to synthetic sickness with Cas9-mediated gene editing events, we utilized a separate CRISPR library targeting genes that encode membrane proteins. Only 19 out of 951 genes (2%) targeted in this library exhibited synthetic lethality with *Polq*[−/−], which is below the 3% false discovery rate threshold used during statistical analyses (Supplementary Fig. 2c, d and Supplementary Data 3). Thus, the large number of PolqSL genes identified in our screen is due to a broad landscape of DDR gene mutations that render cells dependent on *Polq* for viability.

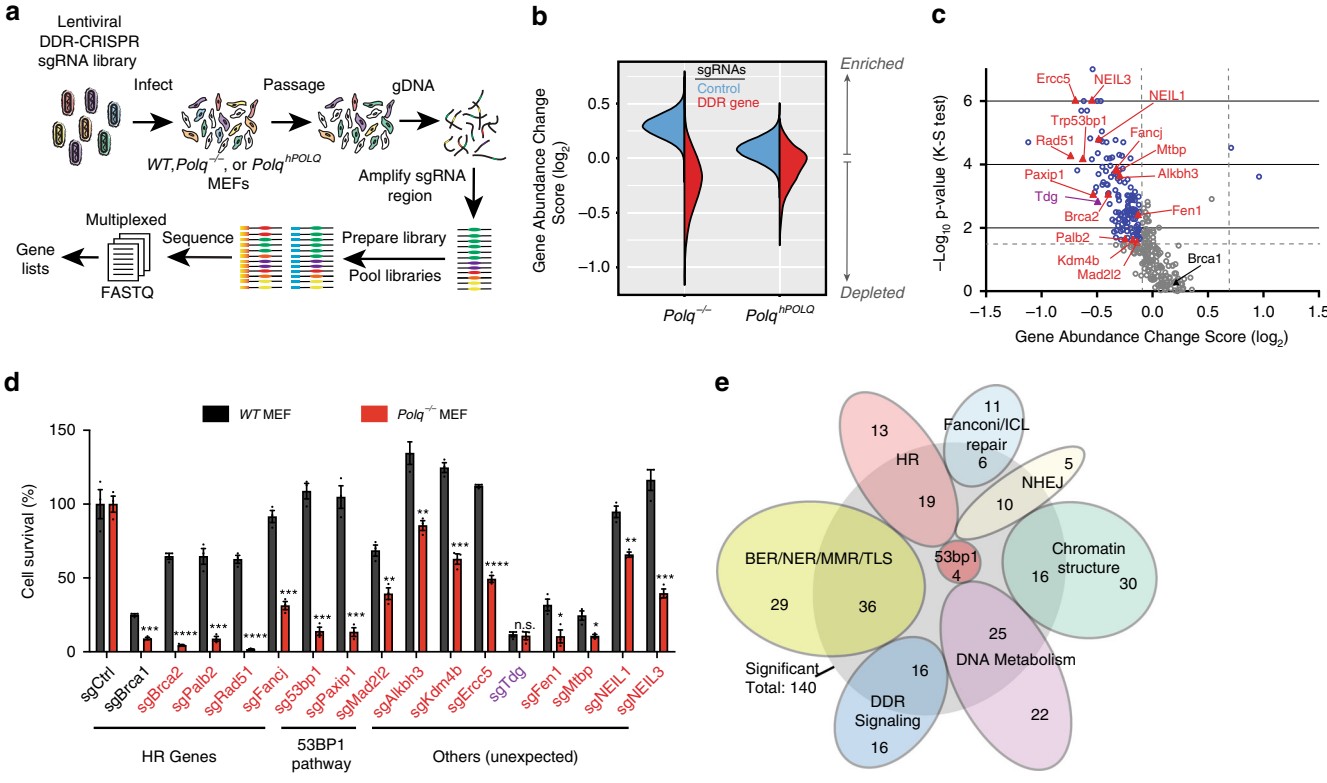

**Fig. 1** Identification of *Polq* synthetic lethal (PolqSL) genes by CRISPR screening. **a** Schematic of the CRISPR genetic screen to identify PolqSL genes. **b** Violin plot of Gene Abundance Change Scores (Log$_2$) for DDR gene-targeting sgRNAs (red) and non-targeting control sgRNAs (blue) in *Polq$^{-/-}$* and *Polq$^{hPOLQ}$* MEFs, relative to *WT* MEFs. **c** Volcano plot of Gene Abundance Change Scores (*Polq$^{-/-}$* versus *Polq$^{hPOLQ}$*) and -Log$_{10}$ p-value of the Kolmogorov-Smirnov test for DDR gene-targeting sgRNAs relative to non-targeting control sgRNAs. Thresholds for statistical significance are indicated by dashed lines (see "Methods" for details). Genes with statistically significant (Blue dots) and non-significant (Gray dots) Gene Abundance Changes Scores are indicated. Genes with red/purple triangles are further validated in Fig. 1d. **d** Relative cell survival measured by colony forming efficiency of *WT* or *Polq$^{-/-}$* cells transduced with a lentivirus containing Cas9 and control sgRNA (sgCtrl) or DDR gene-targeting sgRNAs. Data shown are the mean ± SEM (*n* = 3 biological replicates). Significance determined using an unpaired, two-tailed t-test (\**p* < 0.05; \*\**p* < 0.01; \*\*\**p* < 0.001, \*\*\*\**p* < 0.0001). **e** Functional classification of PolqSL genes identified in our CRISPR screen depicted as a Euler diagram

We validated 15 of the candidate PolqSL genes using standard colony forming assays after transduction with gene-targeting or control sgRNAs (Fig. 1d). For 14 out of 15 genes, we observed significantly reduced viability in *Polq$^{-/-}$* MEFs relative to *WT* MEFs (i.e., 93% hit validation rate). We also tested sgRNAs against *Brca1* in *WT* and *Polq$^{-/-}$* MEFs, due to previously published reports of a synthetic lethal interaction[13,18], although *Brca1* did not emerge as a significant genetic interaction in our CRISPR screen. We observed a modest yet statistically significant reduction in cell viability when sgBrca1 was introduced in *Polq$^{-/-}$* cells relative to *WT* cells. The relatively small magnitude of viability difference between *Polq$^{-/-}$* and *WT* MEFs transduced with sgBrca1 may explain why it was a false negative result in our screen.

Previous work identified two members of the HR pathway (*Brca1* and *Brca2*)[13,18], and 1 member of the NHEJ pathway (*Ku70*)[10], as synthetic lethal with *Polq* deficiency. Our work considerably expands the list of DSB repair genes that are synthetic sick or lethal with *Polq*, such that it now includes 13 additional HR mediators, as well as 4 additional genes specific to NHEJ (Fig. 1e, Supplementary Data 2). We also observed highly significant synthetic sickness between *Polq* deficiency and all four components of the 53BP1 anti-resection pathway included in our screen (*53bp1*, *Paxip1*, *Mad2l2*, and *Rif1*). Surprisingly, many of the remaining PolqSL genes have no direct role in canonical DSB repair. These include genes involved in base/nucleotide excision repair, translesion synthesis, mismatch repair, DNA metabolism, DDR signaling, chromatin structure, and the Fanconi Anemia

repair pathway (Fig. 1e). We postulated that a common feature of these gene mutations may be an increase in endogenously generated replication-associated DSBs. To directly test whether Pol θ is essential for repair of collapsed replication forks, we quantified chromosomal aberrations after Aphidicolin treatment. *Polq$^{-/-}$* MEFs accumulated significantly more metaphase aberrations and had reduced viability after aphidicolin treatment relative to *WT* or *Polq$^{hPOLQ}$* MEFs (Supplementary Fig. 3a-c). Loss of Neil3 has previously been shown to increase replication-associated DSBs[23]. We identified *Neil3* as a PolqSL gene, and observed that CRISPR-mediated knockout of *Neil3* increased nuclear 53BP1 foci more significantly in *Polq$^{-/-}$* MEFs relative to *WT* MEFs, which is consistent with an accumulation of unrepaired replication-associated DSBs. Collectively, these findings argue an essential role for *Polq* is not limited to cells deficient in *BRCA1/2*, or even cells deficient in DSB repair—we show Pol θ is an important compensatory repair mechanism in the background of deficiency in many genes implicated in DDR.

**Synthetic lethality of *Polq/53bp1* DKO cells despite HR and NHEJ proficiency.** Synthetic lethality between *Polq* and *53bp1* has previously been reported[10], and had been presumed to be due to deficiency in NHEJ. To evaluate this possibility, we measured NHEJ, HR, and TMEJ repair at a CRISPR/Cas9 induced break at the murine Rosa26 locus using digital PCR (dPCR) assays designed based on previously published high throughput

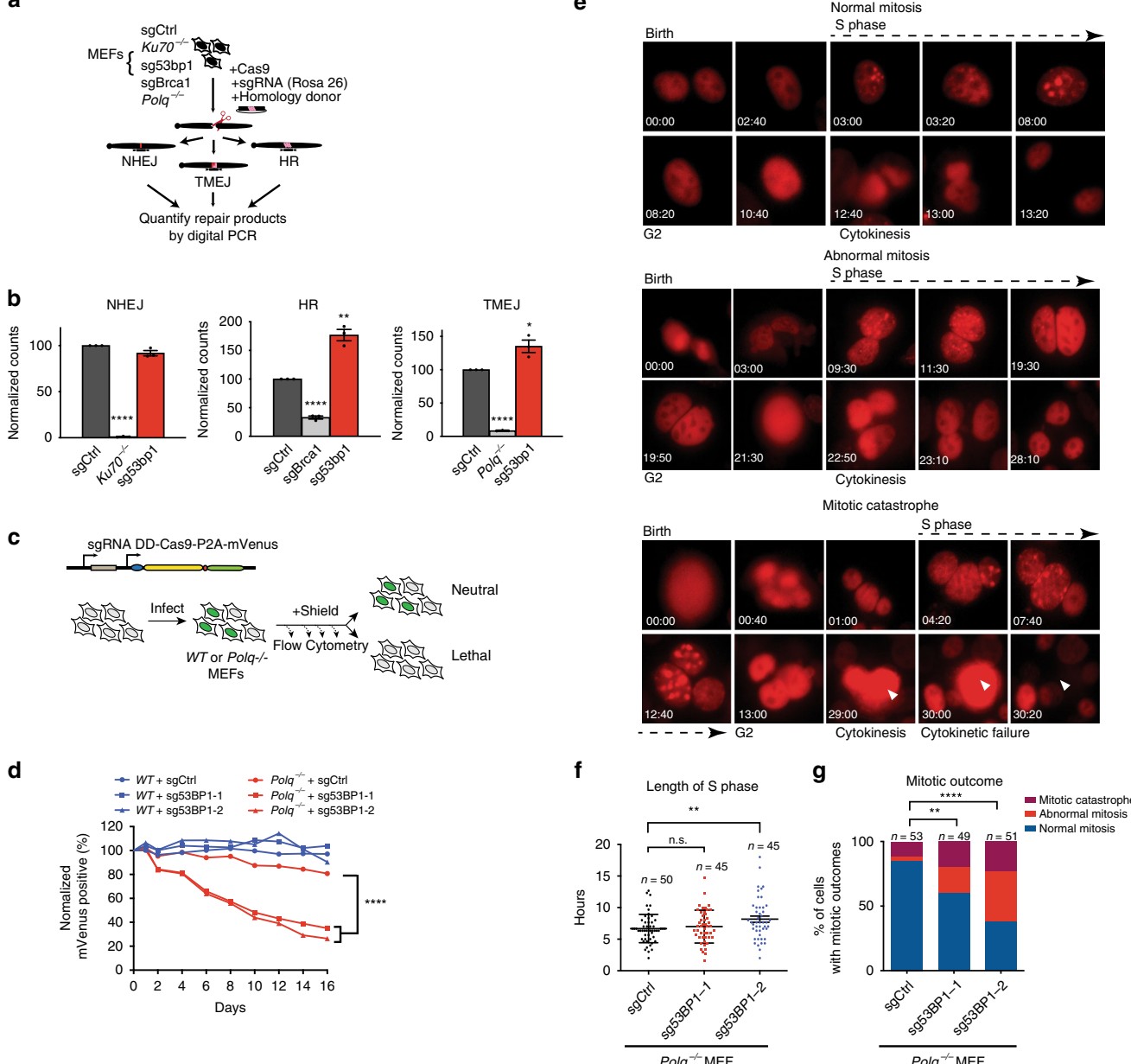

**Fig. 2** Synthetic lethality of *Polq/53bp1* double knockout cells despite HR and NHEJ proficiency. **a** Schema for marker-free quantification of DSB repair pathway choice at a CRISPR-induced chromosomal break using dPCR. **b** Quantification of HR, NHEJ, and TMEJ repair at the *Rosa26* locus, relative to *WT* + sgCtrl. Data shown are the mean ± SEM ($n = 3$ biological replicates). Significance determined using an unpaired, two-tailed *t*-test (*$p < 0.05$; **$p < 0.01$; ****$p < 0.0001$). **c** Diagram of growth competition assay to assess kinetics of synthetic lethality between *Polq* and *53bp1*. DD-Cas9 is protected from degradation upon exposure to a synthetic ligand (Shield1). Depletion of mVenus-positive cells over time after Shield1 exposure is indicative of sgRNA lethality. **d** Normalized percentage of mVenus-positive cells over time after Shield1 treatment, measured by flow cytometry. The mean fraction of mVenus-positive cells ± SEM ($n = 3$) is shown for the various genotypes, normalized to day 0 (no shield). *Polq*$^{-/-}$ + sg53bp1-1 or -2 ****, $p < 0.0001$ using two-tailed nonparametric Spearman correlation in GraphPad Prism v7.04. **e**–**g** Time lapse microscopy of PCNA-mCherry to assess cell cycle phase transitions in individual *Polq*$^{-/-}$ cells 48 h after Shield1 treatment. Cell lines used in this experiment are described in (**c**) after mVenus sorting. **e** Image descriptions for the three mitotic outcomes. Mitotic catastrophe is a terminal event with cells undergoing nuclear degradation, and abnormal mitosis refers to cytokinesis failure or chromosomal mis-segregation events resulting in abnormal nuclear structure in subsequent daughter cells. **f** Distribution of S phase lengths in *Polq*$^{-/-}$ MEFs transduced with sgCtrl, sg53bp1-1, and sg53bp1-2. **$p < 0.01$ using a two-tailed *t*-test. **g** Analysis of mitotic outcome of individual cells with the indicated genotypes. Abnormal mitosis refers to cytokinetic failure or mis-segregation events resulting in abnormal nuclear structure in subsequent daughter cells. Mitotic catastrophe refers to abnormal mitoses resulting in cell death or disappearance. Significance determined using a Chi-square test (**$p < 0.01$ and ****$p < 0.0001$)

sequencing analyses at this locus[10] (Fig. 2a). 53bp1 deficiency resulted in a nonsignificant reduction in NHEJ and increased frequencies of both HR and TMEJ repair (Fig. 2b), consistent with previously described roles for 53bp1 in DSB repair pathway choice[24]. Based on these observations, NHEJ deficiency cannot explain synthetic lethality between 53bp1 and Polq. However, it was also surprising that HR could not compensate for TMEJ deficiency in 53bp1/Polq DKO cells, given that they both act on resected DSBs. We therefore investigated the synthetic lethal phenotype in greater detail.

To assess kinetics of synthetic lethality between 53bp1 and Polq, we utilized an inducible Cas9 expression system (DD-Cas9[25]), and monitored the relative growth rate of transduced cells by flow cytometry over time (Fig. 2c). Interestingly, although 53BP1 expression was already diminished by 48 h after Shield1 treatment (Supplementary Fig. 4), the growth disadvantage of $Polq^{-/-}$ + sg53bp1 (i.e., Polq/53bp1 double knockout, DKO) cells persisted over at least 14 days (Fig. 2d). Time lapse microscopy using PCNA-mCherry as a fluorescent cell cycle reporter[26] revealed a statistically significant prolongation of S phase duration with one out of two 53bp1-targeting sgRNAs in $Polq^{-/-}$ cells (Fig. 2f), although G1 and G2/M duration did not differ significantly for any of the genotypes (Supplementary Fig. 5). More strikingly, there was a significantly higher rate of aberrant mitoses (improper chromosomal segregation or abnormal cytokinesis) and mitotic catastrophe when either of the 53bp1 targeting sgRNAs was expressed in $Polq^{-/-}$ cells (Fig. 2e, g).

**Polq/53bp1 DKO cells accumulate aberrant HR intermediates in S phase.** Because mitotic aberrations can arise from unresolved DNA damage in the preceding S phase[27–29], we performed co-immunofluorescence for Rad51 and γH2AX to assess levels of HR intermediates and DNA damage-associated chromatin, respectively. Notably, we observed large Rad51 aggregates selectively in Polq/53bp1 DKO cells, which frequently were also positive for γH2AX (Fig. 3a, b). The Rad51 foci observed in Polq/53bp1 DKO cells were substantially larger than spontaneous Rad51 foci that occur in a normal S phase in WT cells (Supplementary Fig. 6a–b). In addition, we analyzed EdU incorporation to distinguish non-S phase cells from cells in early, middle, or late S phase (Fig. 3c). The most significant increase in abnormal Rad51 aggregates was observed in middle and late S phase cells (Fig. 3d). We hypothesized that these Rad51 foci arose from spontaneous replication fork collapse. Indeed, aphidicolin treatment increased the percentage of nuclei with large Rad51 foci (Fig. 3e). Furthermore, Rad51 foci that formed in Polq/53bp1 DKO cells persisted even after 12 h, a timepoint when a significant fraction of Rad51 foci had resolved in WT, WT + sg53bp1, and $Polq^{-/-}$ cells (Fig. 3e and Supplementary Fig. 6c). Collectively, these observations indicate that synthetic lethality between 53bp1 and Polq deficiency is due to unsuccessful HR-mediated repair of a subset of replication-associated DSBs.

**Polq is required for Mitomycin C (MMC) induced DNA damage repair.** We next evaluated potential roles for Pol θ after exposure to agents known to cause stalled replication forks, including MMC, which introduces interstrand crosslinks (ICL), and pyridostatin (PDS), which stabilizes G quadruplex (G4) DNA. Prior findings in Drosophila[8,12,30] have implicated Polq in ICL repair. In contrast, MEFs expressing a hypomorphic Polq allele, $Polq^{chaos1}$, were not hypersensitive to MMC[12]. We find that $Polq^{-/-}$ MEFs are hypersensitive to MMC, which can be restored by reconstitution with human POLQ (Supplementary Fig. 7). The discrepancy between these findings may be due to residual activity of the $Polq^{chaos1}$ allele in mediating ICL repair. $Polq^{-/-}$

MEFs exposed to a low dose of MMC (20 ng/mL) had a significantly higher frequency of mitotic crossovers in a sister chromatid exchange (SCE) assay and unrepaired chromosomal aberrations than was observed in wild type cells. Increases in both classes of aberrations were reversed upon exogenous expression of human POLQ (Fig. 4a–c). These observations indicate that TMEJ is a major pathway for ICL repair in mammals that prevents accumulation of mitotic crossovers. Notably, $Polq^{-/-}$ cells treated with MMC also accumulated large Rad51 foci (Fig. 4d–f), similar in character to those observed in Polq and 53bp1 DKO cells.

**Polq is required for pyridostatin induced DNA damage repair.** Pol θ has been implicated in repair of replication-dependent DNA damage at G4 DNA in C. elegans[14,15]. Our CRISPR screen identified a synthetic sickness genetic interaction between Polq and Fancj, which was validated by performing a colony forming assay (Fig. 1d). Fancj is a conserved helicase that unfolds G4 DNA[31] and mutations in its C. elegans ortholog, dog-1, result in high levels of TMEJ signature repair at G4 sites in the genome[32]. We found that $Polq^{-/-}$ MEFs are hypersensitive to the G4 stabilizer pyridostatin (PDS)[33] relative to WT cells (Fig. 5a). Similarly, WT MEFs transduced with sgRNA targeting the Polq polymerase domain induced sensitivity to PDS relative to a control sgRNA (Fig. 5b). $Polq^{-/-}$ cells treated with PDS accumulate a significantly greater number of Rad51 and 53BP1 foci (Fig. 5c–e). Interestingly, Rad51 foci in $Polq^{-/-}$ cells were larger and more frequently adjacent to 53BP1 foci than in WT cells (Supplementary Fig. 8). Altogether, these observations demonstrate an essential role for Pol θ in protection against accumulation of non-productive HR intermediates at sites of replication-associated DNA damage.

**Elevated TMEJ repair signatures in cells with PolqSL gene mutations.** We next investigated whether there was more frequent utilization of TMEJ for DSB repair in cells deficient in genes represented in the PolqSL list. We first induced chromosomal breaks in a wild type MEF line, as well as stable variants of this line deficient in 53bp1 ($53bp1^{-/-}$) or Brca2 ($Brca2^{Mut/-}$) (Supplementary Fig. 9), and characterized repair of these breaks by high throughput sequencing (Fig. 6a). TMEJ events were defined as deletions >5 bp with >2 bp flanking microhomology (MHD), which is a signature pattern of repair product that has previously been shown to be Pol θ-dependent in this cell line[10]. Both of the PolqSL list gene mutants (Brca2 and 53bp1) showed increased use of the TMEJ signature (Fig. 6b, c), although $53bp1^{-/-}$ had longer MHD compared to $Brca2^{Mut/-}$, likely due to increased DSB resection in 53bp1 deficient cells. Similar results were observed using a dPCR assay specific for a Pol θ-dependent MHD in cells that were CRISPR-targeted for two additional HR genes in the PolqSL list, Palb2 and Rad51 (Fig. 6d, e). Frequent synthetic lethality with Polq deficiency thus tightly correlates with the importance of TMEJ as a commonly used compensatory, or backup mechanism for repair of replication-associated DSBs.

**Elevated TMEJ repair signatures in human breast cancers with PolqSL gene alterations.** The association between PolqSL gene mutations and increased utilization of TMEJ repair in MEFs led us to hypothesize that human cancers with PolqSL gene mutations may also contain higher levels of TMEJ-associated genomic scars. Towards this end, we identified 275 out of 926 (29.7%) breast cancers in the TCGA cohort[34] as likely deficient in one or more of the 140 PolqSL genes identified in our CRISPR screen (Supplementary Data 4–5), due to a truncating mutation or a deep copy number deletion. Notably, this is a much larger

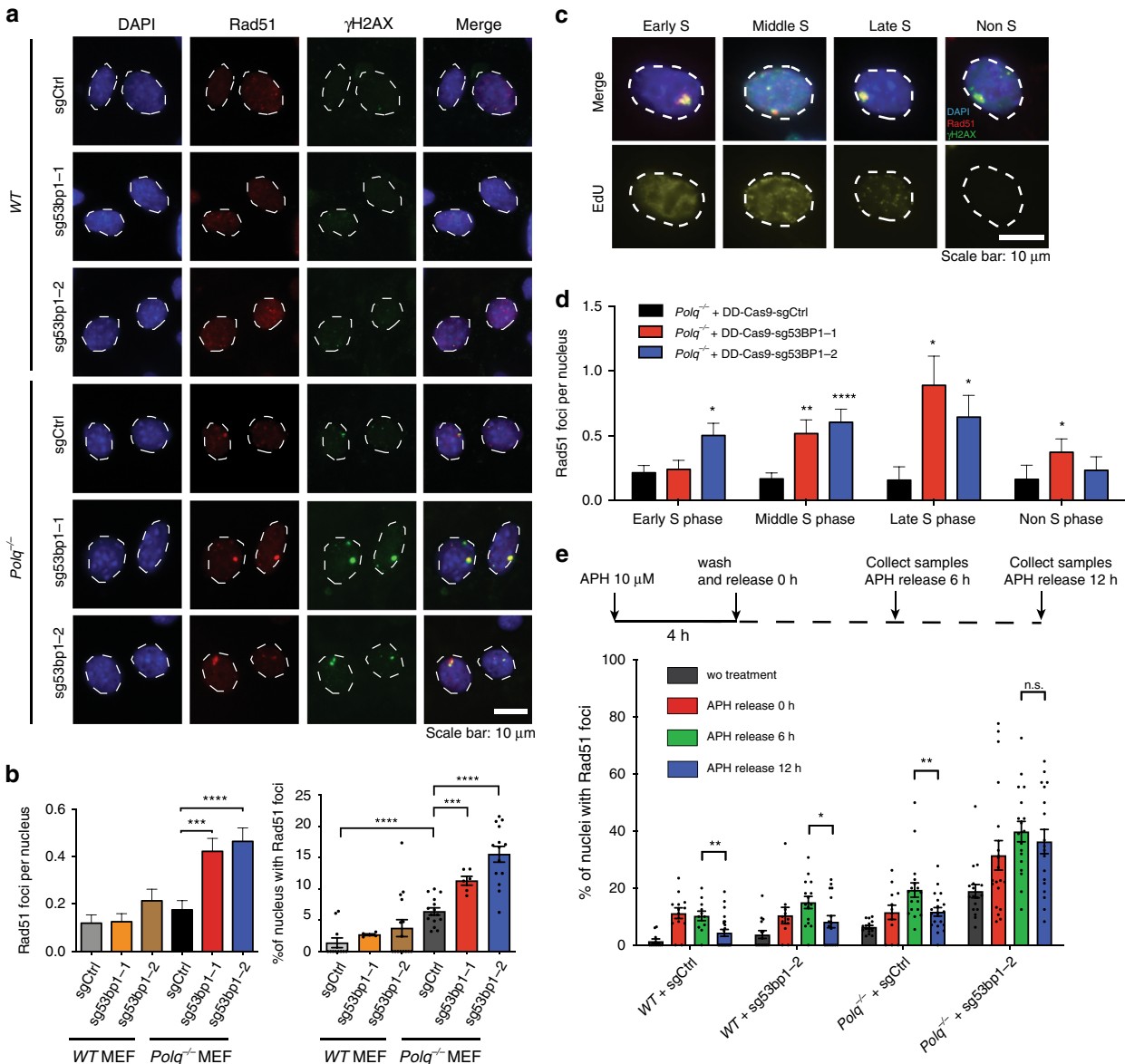

**Fig. 3** *Polq/53bp1* double knockout cells accumulate non-productive HR intermediates in S phase. **a** Representative immunofluorescence (IF) staining of Rad51 (red) and γH2AX (green) foci formation in cells with the indicated genotypes, 48 h after Shield1 treatment. (*n* = 3 biologically independent experiments). **b** Quantification of large Rad51 foci per nucleus and percentage of nucleus with Rad51 foci. Data shown are mean ± SEM, and consistent across three independent biological replicates. ***$p < 0.001$; ****$p < 0.0001$ using a Mann–Whitney test. **c** Representative co-IF images for Rad51 (red), γH2AX (green), and EdU (yellow,10 min EdU pulse) in *Polq⁻/⁻* + sg53BP1-2 MEFs to distinguish cell cycle stages as indicated. **d** Quantification of large Rad51 foci per nucleus stratified by cell cycle stage. *$p < 0.05$; **$p < 0.01$; ****$p < 0.0001$ by a Mann–Whitney test. **e** Cells were treated with 10 μM Aphidicolin for 4 h followed by release for 0, 6, and 12 h, fixed cells and stained cells with Rad51. Quantification of the percentage of nuclei with Rad51 foci was performed. Data shown are mean ± SEM, and consistent across two independent biological replicates. *$p < 0.05$; **$p < 0.01$ using a Mann–Whitney test

fraction of cancers than previously considered as having addiction to Pol θ—only 21 of these 275 cases were *BRCA* mutated. We observed significantly higher levels of *POLQ* mRNA (Fig. 6f) in breast cancers with PolqSL gene alteration. We also investigated correlation with COSMIC mutation signature 3, which is upregulated in cancers with *BRCA1/2* deficiency and also in BRCA non-mutant cancers with suspected homologous recombination deficiency (HRD)[35,36]. We observed highly significant enrichment of COSMIC signature 3 in breast cancers with PolqSL gene alterations (Fig. 6g), relative to breast cancers without PolqSL gene alteration. These observations are consistent with excessive employment of Pol θ in PolqSL deficient cancers. We further explored this possibility by implementing a validated algorithm

for indel detection[37] to quantify the signature readout of TMEJ repair—microhomology-flanked deletions (MHD), defined as deletion size of 5 bp or greater and 2 bp or more of flanking microhomology. Breast cancers with PolqSL gene alterations were significantly more likely to have a detectable TMEJ signature MHD identified from whole exome sequencing (WES) analyses (Fig. 6h). As expected, whole genome sequencing (WGS) identified a 20-fold higher rate of TMEJ signature MHD than WES in a subset of 94 TCGA breast cancers for which both WES and WGS were performed (Fig. 6i, Supplementary Data 6). Forty one out of 94 (43.6%) breast cancers with WGS data available in TCGA had PolqSL gene alterations, and this subset of cancers had significantly higher levels of TMEJ signature MHD than cancers

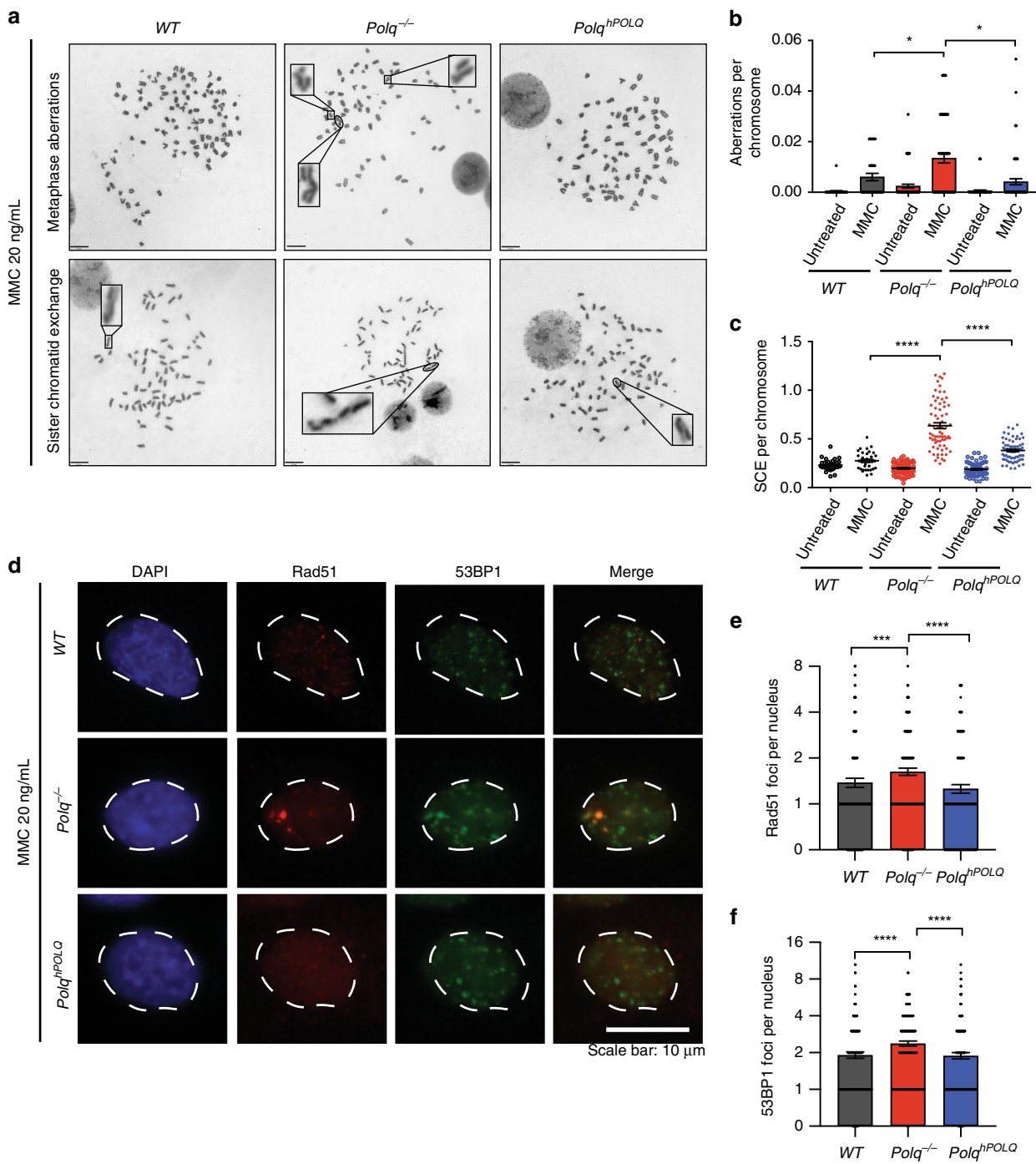

**Fig. 4** *Polq* is required for Mitomycin C induced DNA damage repair. **a** Metaphase aberrations and sister chromatid exchanges are shown in *WT*, *Polq⁻/⁻* and *PolqhPOLQ* cells 12 h after treatment with 20 ng/mL Mitomycin C (MMC). Scale bar = 10 μm. **b, c** Quantification of (**a**), 35 metaphase spreads for each condition was scored, and shown are mean ± SEM. Significance determined using an unpaired, two-tailed *t*-test (*$p < 0.05$; ****$p < 0.0001$). **d** IF analysis of *WT*, *Polq⁻/⁻*, and *PolqhPOLQ* cells six hours after treatment with 20 ng/mL MMC, stained with DAPI (blue) and antibodies specific for Rad51 (red) and 53BP1 (green) ($n = 3$ biologically independent experiments). (**e**, **f**) are quantification of (**d**). Statistical significance was assessed by unpaired, two-tailed Mann–Whitney tests (***$p < 0.001$; ****$p < 0.0001$)

without PolqSL gene alterations (Fig. 6j). Thus, mimicking our findings in genetically engineered MEFs, we find that human breast cancers with deficiency in PolqSL genes have multiple indices of a hyperactive TMEJ repair pathway.

## Discussion
We have defined a surprisingly diverse landscape of DDR gene mutations that renders cells addicted to TMEJ for survival. The functional diversity of PolqSL genes suggests that Pol θ becomes essential upon increased levels of endogenous, unrepaired DNA damage, regardless of the precise nature of that damage. The lack of specificity for a specific type of DNA damage argues against a translesion synthesis function for Pol θ, and is consistent with Pol θ-mediated repair of replication-associated DSBs via TMEJ. Indeed, we found Pol θ is essential for repair of DSBs arising from aphidicolin-induced replication fork collapse (Supplementary Fig. 3).

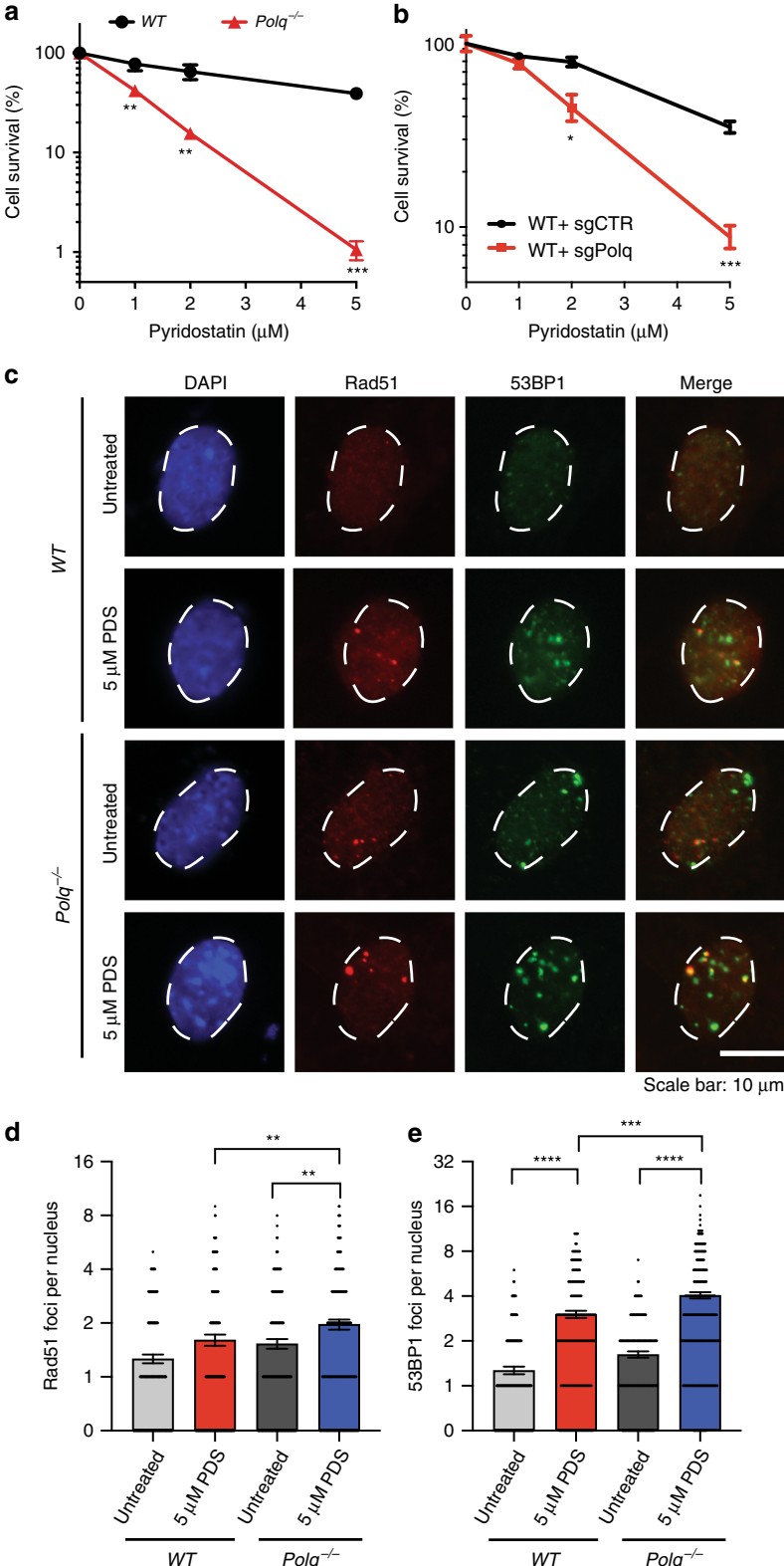

**Fig. 5** *Polq* is required for Pyridostatin induced DNA damage repair. **a** Colony forming efficiency after treatment with Pyridostatin (PDS, 1, 2, 5 μM) in *WT* and *Polq*−/− MEFs. **b** Colony forming efficiency after treatment with Pyridostatin (PDS, 1, 2, 5 μM) in *WT* MEFs transduced with Cas9 and either sgCtrl or sgPolq (targeting the polymerase domain). **a**, **b** Data shown are the mean ± SEM (*n* = 3). Statistical significance was assessed by two-tailed *t*-tests. *$p < 0.05$, **$p < 0.01$ and ***$p < 0.001$. **c** IF images for DAPI (blue), Rad51 (red) and 53BP1 (green) in *WT* and *Polq*−/− MEFs six hours after treatment with 5 μM PDS (*n* = 3 biologically independent experiments). **d**, **e** are quantification of large Rad51 and 53BP1 foci, as observed in (**c**). Shown are mean ± SEM, and representative of three independent experimental replicates. Statistical significance was assessed by Mann–Whitney tests. **$p < 0.01$, ***$p < 0.001$, and ****$p < 0.0001$

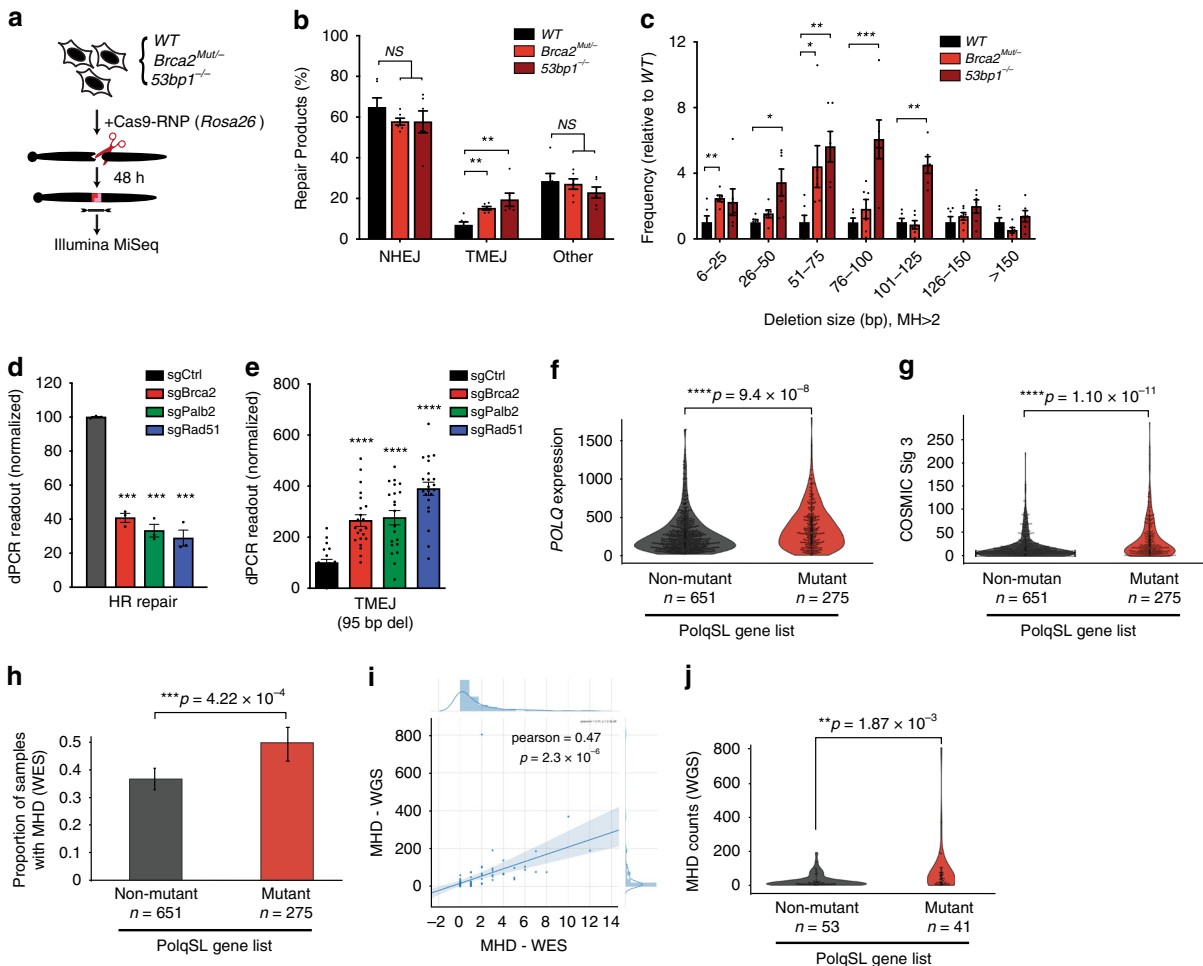

**Fig. 6** Elevated TMEJ repair signatures in cells and cancers with PolqSL gene mutations. **a** *WT*, *53bp1*$^{-/-}$, and *Brca2*$^{Mut/-}$ MEFs were transfected with Cas9 ribonucleoprotein (Cas9-RNP) targeting the Rosa26 locus. Forty-eight hours later the targeted region was amplified and analyzed by high throughput sequencing (Illumina MiSeq). **b** Percentage of repair products classified as NHEJ (≤5 bp del, or 1–3 bp insertion), TMEJ (>5 bp deletion and >2 bp MH), and "Other" (>5 bp deletion and 0–2 bp MH), for the indicated MEF genotypes. **c** The relative frequency (normalized to *WT*) of end joining products with >2 bp MH and deletion size within the indicated ranges. *53bp1*$^{-/-}$ MEFs are associated with larger-sized TMEJ signature deletions. **b**, **c** Mean values ± SEM ($n =$ 6) are shown. Significance determined using an unpaired, two-tailed *t*-test (*$p < 0.05$; **$p < 0.01$; ***$p < 0.001$). **d**, **e** DNA repair products are detected by digital PCR using WT MEFs transduced with the indicated PolqSL gene-targeting sgRNA. HR is detected after co-transfection of a homology donor. Relative rates (normalized to *WT* + sgCtrl) of (**d**) homologous recombination (HR) and (**e**) TMEJ (95 bp deletion with 5 bp MH) are indicated as mean values ± SEM ($n = 3$). Significance determined using an unpaired, two-tailed *t*-test (***$p < 0.001$; ****$p < 0.0001$). **f–j** Analysis of breast cancers in TCGA. PolqSL mutant breast cancers are identified using cBioPortal as having a truncating mutation or deep copy number deletion. *POLQ* mRNA expression (**f**) and COSMIC Signature 3 (**g**) are elevated in cancers with PolqSL gene alterations. **h–j** Whole exome sequencing (WES) and whole genome sequencing (WGS) analyses in TCGA breast cancers to detect MH-flanked deletions (MHD). **h** Breast cancers with PolqSL gene alterations are more likely to have a detectable MHD by WES than breast cancers without alteration in PolqSL genes. The error bar is a bootstrapped 95% confidence interval. WGS data is available for 94 TCGA breast cancers. **i** High correlation between MHD detected by WGS or WES. **j** Significantly higher MHD count, detected by WGS, among breast cancers with PolqSL gene alterations, relative to the non-altered breast cancers. **f–g**, **i–j** Statistical significance was assessed by two-tailed Mann–Whitney tests. *$p < 0.05$, **$p < 0.01$, ***$p < 0.001$, and ****$p < 0.0001$

Whilst prior studies suggested that TMEJ primarily functions as a backup pathway to HR and NHEJ, our study identifies numerous examples of TMEJ essentiality when canonical DSB repair pathways are unperturbed. Analysis of synthetic lethality in *53bp1/Polq* DKO cells reveals an accumulation of unrepaired HR intermediates in S phase that is further exacerbated by aphidicolin-induced replication fork collapse. We also find TMEJ essentiality upon G quadruplex stabilization and after exposure to interstrand crosslinking agents, both of which promote replication fork stalling and/or collapse. These observations suggest that TMEJ is required for repair of a subset of replication-associated DSBs that is not amenable to repair by HR (Fig. 7). An example of such a break could be one where the template contains a

replication-blocking lesion. Recent evidence supports a model wherein unresolved replication-blocking lesions can be inherited as tracts of single-stranded gaps surrounding the lesion[15,38,39]. Replication of these lesions in daughter cells would give rise to two-ended DSBs that are not amenable to HR due to persistence of the replication-blocking lesion in the template DNA strand (Fig. 7, left panel). TMEJ may be a preferred repair mechanism at these sites due to its ability to re-join resected breaks without requiring a homologous template (Fig. 7, right panel). Thus, we postulate that DDR gene mutations that induce a higher prevalence of unresolved replication-blocking lesions may induce TMEJ essentiality. Alternative activities of Pol θ may also be operative at replication-associated DSBs. For example, prior

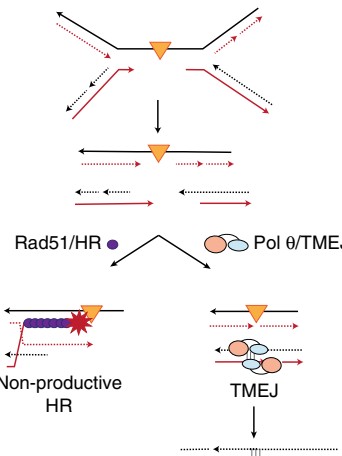

**Fig. 7** Model for TMEJ in suppressing non-productive HR at replication-associated DNA damage. Orange triangle indicates a replication-obstructing lesion, such as ICL, G4, or base damage, with an associated region of under-replicated DNA. Converging replication forks will generate a two-ended DSB that can undergo end resection to expose 3′ overhangs. Rad51 loading and attempted HR may result in unsuccessful repair due to persistence of the replication blocking lesion in the homologous template DNA. Alternatively, TMEJ is able to perform microhomology-mediated end joining of the exposed 3′ overhangs, without requiring a homologous template

studies have shown that Pol θ can promote microhomology-mediated integration of plasmid DNA, implying an ability to invade donor templates that lack overt DSBs[40,41]. Future studies will be necessary to unravel the mechanism by which Pol θ resolves stalled or collapsed replication forks, and its relationship to HR-mediated repair.

Our findings also demonstrate that a hallmark feature of cells and cancers with gene mutations that induce Pol θ addiction is an increased prevalence of TMEJ pattern genomic scars. The striking similarity and overlap between genomic scar signatures previously ascribed to HR deficiency (HRD) or "BRCA-ness"[42,43] and TMEJ repair raises the distinct possibility that hyperactive TMEJ is the etiologic driver of this genomic scar pattern in human cancer. While we have shown that HRD induces hyperactive TMEJ, our study also demonstrates that non-HR gene alterations (such as 53BP1) are also sufficient to induce TMEJ hyperactivity and addiction. An important clinical implication of these findings is that only a subset of cancers identified by HRD or BRCA-ness genomic scar signatures may be functionally HR deficient. This may explain the incomplete correlation between HRD signatures and functional HR assays[44], and the recent finding that HRD signatures are unable to accurately predict platinum chemotherapy sensitivity in metastatic breast cancer patients[45]. We propose that genomic classifiers that incorporate COSMIC signature 3, and especially MHD burden, may be more precisely described as "hyper-TMEJ" signatures, rather than signatures specific for HRD.

Our study suggests that cancers with "hyper-TMEJ" signatures may be dependent on Pol θ for their survival. Recent pan-cancer genomic analyses[46,47] suggest that hyper-TMEJ signatures and PolqSL gene deficiency may account for as many as 20% of all human cancers, thus greatly expanding the number of cancers for which Pol θ represents an attractive therapeutic target. Refining the optimal "hyper-TMEJ" genomic scar signature that predicts Pol θ addiction in human cancers will be a clinically relevant area of future investigation.

## Methods

**Cell culture**. WT (Polq$^{+/+}$), Polq$^{-/-}$, Polm$^{-/-}$, and Polm$^{+/+}$ cells were SV-40 large T antigen immortalized MEFs[12,20,22], while Polq$^{hPOLQ}$ MEFs were generated by complemented Polq$^{-/-}$ MEFs by human POLQ cDNA expression[20]. 293T cells were purchased from ATCC (CRL11268). All cells were maintained in Dulbecco's modified Eagle's medium (DMEM), with 10% Bovine Calf Serum (Hyclone BCS) and 2 mM L-glutamine (ThermoFisher). Polq$^{hPOLQ}$ cells were maintained in the same media supplemented with 2 μg/ml puromycin. All cells were maintained at 37 °C in an atmosphere of 5% $CO_2$. Cells in culture were routinely monitored for mycoplasma contamination using the Plasmo Test™ (Invivogen).

**Oligo synthesis and pooled library cloning**. DNA oligonucleotide sgGuide library was synthesized by LC Sciences (Supplementary Data 1). A subset of this library was then amplified by PCR using AmpliTaq Gold® 360 DNA Polymerase (ThermoFisher) with forward primer ArrayF and reverse primer ArrayR (Supplementary Data 7) followed by purification with MinElute PCR Purification Kit (Qiagen) to produce a double strand product suitable for Gibson cloning[48]. The CRISPR library cassette was cloned into lentiCRISPR v2 (a gift from Feng Zhang, Addgene plasmid # 52961) followed by transformation into Endura™ Electro-Competent Cells (Lucigen) according to the manufacturer's protocol using BTX Gemini system (ThermoFisher). To ensure no loss of representation, six parallel transformations were performed using the same Gibson reaction and plated into twelve, 10 cm petri dishes (VWR) containing LB agar (ThermoFisher) with 100 μg/ml carbenicillin (ThermoFisher). Colonies were scraped off plates and combined for DNA extraction (Qiagen).

**Lentivirus generation**. Lentiviruses were generated by 293T cells in 150 mm dish with transfection of 3 μg pMD2.G (a gift from Didier Trono, Addgene # 12259), 4.5 μg psPAX2 (a gift from Didier Trono, Addgene # 12260) and 6 μg custom DDR-CRISPR pooled lentiviral library, transfection was performed using Poly-ethylenimine (Linear, MW 25,000, Polysciences, Inc)[49,50]. Supernatant from the packaging reaction was collected at 48 h and 72 h. This was pooled and then filtered through 0.45 μm filter. The virus was then concentrated by pelleting at 113,000 × $g$ in a SW28 ultracentrifuge rotor for two hours at 4 °C. The pellet was allowed to dissolve overnight in desired volume of PBS at 4 °C and then aliquoted and frozen at −80 °C.

**CRISPR library screening**. Our custom DDR-CRISPR pooled lentiviral plasmid library containing 3908 sgRNAs targeting 309 murine DNA damage response (DDR) genes with an average of 10 sgRNAs per gene, as well as 834 non-targeting sgRNA controls was used to infect cells at a MOI ~0.8. Twenty-four hours after addition of virus, the media was removed and replaced with fresh media. Forty-eight hours after adding the virus the cells were split. One million cells from each infection was seeded into a 15 cm dish in media containing 2 μg/ml puromycin. Cells were passaged once every two to three days, 1 × 10$^6$ cells were reseeded into a 15 cm plate each time. After 8 population doublings, cells were harvested and genomic DNA isolated using QIAamp DNA Blood Kit (Qiagen). To amplify lentiCRISPRv2 sgRNAs, PCR was performed in two steps: For the first PCR, the amount of input genomic DNA (gDNA) for each sample was calculated to achieve 120 X coverage over the DDR-CRISPR library, which resulted in 5 μg DNA per sample (assuming 6.6 μg of gDNA for 10$^6$ cells). For each sample 5 separate 100 μl PCR reactions with 1 μg genomic DNA in each reaction using Herculase II Fusion DNA Polymerase (Agilent) were carried out using DDR_CRISPR_Ion_1$^{st}$_FWD and DDR_CRISPR_Ion_1$^{st}$_REV and then combined. A second PCR was performed to attach Ion adaptors and to barcode samples. The second PCR was done in two 50 μl reactions using around 80 ng product from the first PCR. Primer sequences for the first and second PCR are attached in Supplementary Data 7. Amplification was carried out with 30 cycles for the first PCR and 10 cycles for the second PCR. Twenty-four to twenty-eight libraries were then pooled and sequenced on an Ion S5 (ThermoFisher) using the 530v1 chip.

The "membrane CRISPR library" was obtained from Addgene (ID: 1000000124), and targets 951 genes that encode membrane associated proteins[51]. Pooled CRISPR library lentivirus was transduced into WT and Polq$^{-/-}$ cells expressing Shield1-inducible DD-Cas9-mVenus. We sorted at least 2 × 10$^6$ cells expressing both DD-Cas9 (mVenus) and membrane CRISPR library (mCherry) before adding Shield1 treatment. 24 h after Shield1, cells were passaged once every two to three days, by reseeding 1 × 10$^6$ cells into a 15 cm plate each time. We performed the same genomic DNA extraction and library prep and Ion sequencing as our DDR CRISPR library, except the first round primers for membrane CRISPR library amplification were: MMB_CRISPR_Ion_1st_FWD and MMB_CRISPR_Ion_1st_REV (Supplementary Data 7).

**CRISPR library analysis**. The number and significance of guides present for each library in the multiplexed FASTQ file is determined using our custom algorithm (Völundr). Völundr identifies and counts the sgRNA sequence in the FASTQ reads allowing for a single mismatch in the sequence. It then writes a count file for each library in the pool and a summary file describing the FASTQ file and the libraries. The count files are used by the Völundr target analysis module as the input data for determining which genes are significantly different than the biological control

sample. To accomplish this the counts in each file are first normalized to the total counts for its library (Supplementary Note 1 see Eq. 1). The normalized data from the replicate samples are then combined on a per guide basis by determining the geometric mean for each guide across replicates (Supplementary Note 1 see Eq. 2). At this step the sgRNA $TD_{Norm}$ value and sgRNA Abundance Change Scores are determined as shown in Supplementary Note 1 see Eq. 3. Any guides with no counts in the Plasmid sample are masked from all analysis at this point. The "Gene Abundance Change Scores" are determined as in Supplementary Note 1 see Eq. 4. For each sgRNA targeting a gene of interest, "ABC", the experimental sample sgGuide $TD_{Norm}$ value is first subtracted from the corresponding biological sample control sgGuide $TD_{Norm}$ value. The $log_2$ transformed, geometric mean of this set is the "Gene Abundance Change Scores". These scores are also computed in the next section on the control guides to empirically estimate the distribution of the Gene Abundance Change Scores in the absence of real biological change using a resampling-based scheme.

The Völundr pipeline takes two different approaches to determine if a targeted gene is significantly different than the biological control. The first is to estimate an empirical null distribution for the Gene Abundance Change Scores by randomly sampling ten non-targeting guides (sgControl) sgRNA $TD_{Norm}$ values, from the 834 sgControl guides a total of 100,000 times (Python NumPy, random choice). For each random set of control guides sampled, we repeat the procedure of the prior paragraph to calculate an empirical Gene Abundance Change Score. The 99.99 percentile and 0.01 percentile values are used as the boundaries for the null set. The second method uses the Kolmogorov-Smirnov test to determine if the sgRNA Abundance Change Scores for the genes were drawn from the same population sgRNA Abundance Change Scores for the sgControls. The p-values of the Kolmogorov-Smirnov test (Python Scipy Stats, ks_2samp) are corrected for multiple tests using a false discovery of 3% with the two-stage linear step-up procedure of Benjamini, Krieger and Yekutieli found in GraphPad Prisim v7.04. The first test evaluates the observed score distribution across the 10 guides for a gene relative to the empirical null distribution across randomly sampled sets of 10 control guides. The second score evaluates the overall gene sgRNA Abundance Change Score relative to the overall sgControl sgRNA Abundance Change Score. For stringency, we require genes to pass both tests to be reported as significant.

**Establishment of mammalian expression constructs and stable cell lines.** DNA corresponding to sgRNAs was cloned into LentiCRISPRv2 (a gift from Feng Zhang, Addgene # 52961), or DD-Cas9 (a gift from Raffaella Sordella, Addgene plasmid # 90085), or pGL3-U6-sgRNA-PGK-puromycin (A gift of Xingxu Huang, Addgene # 51133), using the same protocol described above. Cells were incubated with fresh lentivirus for 24 h and then were recovered for another 24 h. Infected cells are selected by 2 μg/μl puromycin or mVenus by flow cytometry.

For 53BP1 and Brca2 mutant cell lines, we used the Alt-R CRISPR-Cas9 system (IDT). We performed transfection using the Neon transfection kit (Invitrogen) according to manufacturer's protocol. Alt-R HiFi Cas9 nuclease, crRNA and tracrRNA were purchased from IDT and were used at the manufacturer's recommended concentration; crRNA is designed using MIT CRISPR (http://crispr.mit.edu). Forty-eight hours after transfection, cells were seeded for single clone outgrowth, PCR screening and Sanger sequencing to confirm gene targeting, and subsequent functional tests.

**Synthetic lethal and colony formation assay.** Two individual sgRNAs were chosen to target each gene. Forty-eight hours after infection, infected cells were selected by puromycin and counted for colony formation efficiency. Cells were incubated for 7–10 days at 37 °C to allow colony formation. Colonies were stained by Coomassie blue. sgRNA sequences are attached in Supplementary Data 7.

**Competitive growth assay.** DD-Cas9-sgRNA was transduced into the indicated cell lines by lentivirus infection. Forty-eight hours later, mVenus positive cells were quantified by flow cytometry. Normally, more than 50% cells are mVenus positive cells. Cells were treated with 200 nM Shield1 (Takara) and were collected for flow cytometry at the indicated time points.

**Time-Lapse microscopy.** Cells stably expressing Proliferating Cell Nuclear Antigen (PCNA)-mCherry were transduced with DD-Cas9-sgRNA. PCNA-mCherry fusion reporter is a gift from Dr. Jeremy Purvis and Hui Chao Xiao. Cells were plated on Cell-Tak (Corning) coated glass-bottom 12-well plates (Cellvis) with Phenol-free DMEM (Invitrogen) supplemented with 10% FBS, and L-glutamine with or without Shield1. Forty-eight hours post plating, cells were image captured every 20 min for 72 h in the mCherry and mVenus fluorescence channels. Fluorescence images were obtained using a Nikon Ti Eclipse inverted microscope with a 40x objective and Nikon Perfect Focus (PFS) system to maintain focus during acquisition period. Cells were maintained at constant temperature (37 °C) and atmosphere (5% $CO_2$). Image analysis was performed on ImageJ – Fiji.

**DNA repair assay.** Cell lines used in the assay are indicated in the figure. $2 \times 10^6$ cells were transfected with 5 μg pGL3-U6-sgRNA-PGK-puromycin (A gift of Xingxu Huang, Addgene # 51133), 5 μg Flag-Cas9 (A gift of Xingxu Huang, Addgene # 44758), with or without 10 μg HR long donor[10] and 1 μg pEGFP-N2

(Takara) by Neon transfection kit (Invitrogen) using a 1350 V, 30 ms pulse in a 100 μL chamber. Forty-eight hours post transfection, a portion of the cells were analyzed by flow cytometry to quantify the transfection efficiency, and the remaining cells were harvested for genomic DNA extraction (Qiagen). Digital PCR (QX-200, Bio-Rad) was performed to quantify the frequency of gene conversion events using the primers and Taqman probes listed in Supplementary Data 7. The repair signal was normalized to 5000 copies of genomic DNA, measured using a Chromosome 6 control dPCR assay, using primers/probes sequences listed in Supplementary Data 7. Analysis of dPCR data was performed using QuantaSoft (Bio-Rad).

**Immunofluorescence.** Cells were permeabilized by CSK buffer (10 mM Hepes, 300 mM Sucrose, 100 mM NaCl, 3 mM MgCl2, and 0.5% Triton X-100, pH = 7.4) for 2 min followed by fixation for 15 min in 3% paraformaldehyde. Cells were subsequently processed for immunostaining experiments using the indicated antibodies. Nuclei were visualized by staining with DAPI. The primary antibodies used were: Rad51 (1:500, Novus Biologicals, NB100-148), γH2AX (1:500, Trevigen, 4418-APC-100), and 53BP1 (1:500 for immunofluorescence, 1:5000 for western blot, Bethyl, A300-272A). The secondary antibodies were: Rhodamine Goat Anti-Mouse IgG (H + L) (1:500, Jackson ImmunoResearch, 115-025-146) and FITC Goat Anti Rabbit IgG (H + L) (1:500, Jackson ImmunoResearch, 111-095-144). For S phase stain, we incubated cells with 10 μM EdU for 10 min, EdU was detected according to the EdU-Click 647 kit protocol (baseclick). Images were acquired using an Olympus BX61 fluorescence microscope or Zeiss 880 with Airyscan processing for the super-resolution images.

**Metaphase and sister chromatin exchange assay.** Metaphases were prepared by a previously published method[52] with the noted changes. Cells were treated with 100 ng/ml of Colcemid (KaryoMAX® 15210-040 from Gibco) for 1 h prior to harvest and swelling in 75 mM potassium chloride for 20 min at 37 °C. Once the metaphases were dropped onto slides, the slides were stored at room temperature for at least two days prior to staining with Giemsa (KaryoMAX® 10092-013 from Gibco) for 2–3 min. After staining, the slides are rinsed with distilled water and allowed to air dry completely before mounting the coverslips with DPX Mountant (Millipore Sigma). Spreads were imaged under a 100× objective using an Olympus BX61 Light Microscope with QImaging RETIGA 4000R camera.

The SCE assay was performed as previously described[53]. Briefly, 24 h after cells were plated, 10 μM bromodeoxyuridine (BrdU) (Millipore-Sigma) was added to the plates for 24 h. MMC (20 ng/ml, Millipore-Sigma) was added for the final 12 h in BrdU. For the final hour 100 ng/ml of Colcemid (KaryoMAX® 15210-040 from Gibco) was added to the media. The cells were harvested by trypsinization and processed for metaphase spreads as described above. After 2–3 days the metaphases were stained for 30 min by placing the slides in a Coplin jar containing 10 μg/ml Hoechst 33342 (Thermo Fisher) in PBS. The slides were then removed from the Hoechst solution and placed in a tray of 2 × SSC (20 × SSC Stock: 3 M sodium chloride with 300 mM sodium citrate) on a 45 °C heat block. While in the warm SSC the metaphases where exposed to UVB radiation from a Danmar UVB compact fluorescent bulb (peak emission 365 nM) at a distance of 5 cm for 20 min. After exposure to UVB place the slides in a Coplin jar of 2 × SSC for ≥10 min to let the Hoechst and degraded DNA wash away. The slides were then stained with Giemsa (KaryoMAX® 10092-013 from Gibco) for 5 min. The slides were cover slipped and imaged as described above for metaphases.

**High throughput sequencing.** Two hundred nanograms of genomic DNA were amplified using a two-step PCR that added unique library bar-codes, heterogeneity spacers and Illumina MiSeq adapters (as in[54]). Two-step PCR primers are attached in Supplementary Data 7. Samples were sequenced using a 2 × 300 MiSeq kit[55]. Quantification and classification of the sequences was done in R and excel.

**Analysis of MHD in human breast cancers.** TCGA WES reads were aligned using bwa-mem (Li, 2013, arXiv:1303.3997 [q-bio.GN])[37]. Read duplicates were marked by bio-bambam2 (https://github.com/gt1/biobambam2[56],). Variants were called with Strelka[57], UNCeqR[58] and Cadabra[37] (https://cadabra.science). Variant calls were annotated with the Variant Effect Predictor (VEP)[59]. For TCGA cases where WGS data were available, reads were trimmed using SeqPurge[60], aligned using bwa-mem and realigned using ABRA2. Read duplicates were marked by biobambam2 and indels were called using Cadabra. Variant calls were annotated with VEP. TCGA mRNA reads were aligned to human reference genome hg38 using STAR[61] and quantified with Salmon[62] which was run against STAR's transcriptome alignments. Quantification values were upper quantile normalized.

MHD were defined as deletions ≥ 5 bp in length with ≥ 2 bp of flanking microhomology. To determine flanking microhomology, the 3′ end of the deletion was matched to the sequence directly upstream of the deletion junction. Deletions located in regions enriched in short repeats were ignored. MHD in TCGA breast cancer samples were summed per sample with a sample being classified as having MHD if at least one occurrence of MHD was observed.

PolqSL mutant TCGA samples were identified as having at least one deep copy number deletion or truncating mutation in a PolqSL gene using cBioPortal. POLQ

mRNA expression, COSMIC mutation signature 3 scores[63], and proportion of samples with MHD were compared between PolqSL mutant and non-mutant groups using a two-tailed Mann–Whitney test.

**Reporting summary**. Further information on research design is available in the Nature Research Reporting Summary linked to this article.

## Data availability

Sequencing data is available at [https://www.ncbi.nlm.nih.gov/sra/PRJNA556352]. Unanalyzed raw data is available at [https://figshare.com/projects/Genetic_Determinants_of_Cellular_Addiction_to_DNA_Polymerase_Theta/67331]. All data is available from the corresponding author upon request. The source data underlying Figs. 1d, 2b, d, f, g, 3b, d, e, 4b, c, e, f, 5a, b, d, e, 6b–e and Supplementary Figs. 1, 3, 5, 6, 7a and 8 are provided as a source data file.

## Code availability

Code for Völundr has been made publicly available at [https://github.com/pkMyt1/Volundr]. Other software for statistical analysis is publicly available and referenced as noted.

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

## Acknowledgements

This work is dedicated to the memory of Alexander Kenan, who was instrumental in the design of the DDR-CRISPR library. We thank S. Kumar, H.X. Chao, M. Kapustina, V. Roberts, and S. Connor for technical assistance and/or data analysis support. We are grateful to Gupta and Ramsden Lab members for helpful discussions. UNC Core labs (Microscopy Services Laboratory, Hooker Imaging Core, Flow Cytometry Core Facility, and High Throughput Sequencing Facility) used in this study are supported in part by P30 CA016086 Cancer Center Core Support Grant to the UNC Lineberger Comprehensive Cancer Center. G.P.G. holds a Career Award for Medical Scientists from the Burroughs Wellcome Fund. Additional funding support was provided by the University Cancer Research Fund (G.P.G., D.A.R.), NCI/NIH (CA222092, D.A.R. and G.P.G., CA193124, R.D.W.), Dept of Defense (W81XWH-18-1-0047, G.P.G. and D.A.R.), the Grady F. Saunders, PhD Distinguished Research Professorship (R.D.W.) and NCI P30 CA016086 (J.S.P.).

## Author contributions

G.P.G. conceived and supervised the study. G.P.G., D.A.R. and W.F. designed the experiments. R.D.W. provided critical reagents and project guidance. W.F. performed the CRISPR screens, D.A.S. developed and implemented the bioinformatics analysis pipeline, and N.R. provided statistical input and review. Unless otherwise stated, W.F. performed all additional experiments and data analyses, with statistical review by N.R. R.J.K. performed live cell imaging experiments with guidance from J.E.P. D.A.S. performed mitotic aberration and SCE experiments. J.C.G. performed Rosa26 high throughput sequencing and data analysis with supervision from D.A.R. B.A.P. and L.E.M. performed analyses of human breast cancer genomic datasets with supervision from J.S.P. G.P.G., D.A.S., W.F. and D.A.R. wrote the manuscript, with contributions from all authors. All authors read and accepted the manuscript.

## Additional information

**Competing interests:** The authors declare no competing interests.

