## [Peer Review File · Nature Communications]

Reviewers' comments:

Reviewer #1 (Remarks to the Author):

In this manuscript, there are some results that will be of interest to the field. However, it would have been even more interesting had a more thorough examination of newly discovered PolqSL interactions been performed. For example, at the outset, it is not clear why so much effort is made to examine the 53BP1 SL, since this same group already reported this SL in an earlier study [Wyatt et al., (2016)]. In addition, this 53BP1 SL excursion does not help with the logical flow from the study of SLs with theta to looking for theta signatures in TCGA breast cancer lines (the last section of the ms. and perhaps the most interesting). As the ms. stands, first, the old 53BP1 SL interaction is re-examined, then some experiments with MMC are introduced and next, yet another drug, PDS is examined for its effect in a Polq^{-/-} background. All of these results are presented before the authors get to the cancer studies. In fact, before the cancer studies are presented, the authors shift to two more genes (Palb2 and Rad51), both of which again were also previously known to show SL with Polq. Overall, the fact that only a very few new PolqSL interactions are examined is a little disconcerting as it impacts the overall novelty of the work. As presented, the ms. appears to be cobbled together from several disparate pieces of work. It certainly would have helped if a few new SL interactions were explored in greater detail, or had there been more emphasis on the kinds of TMEJ events that are found in TCGA.

On another note, in the cancer section on Ins. 230-233: Is 30% actually a high number? If one were to take 309 genes from another non-essential process, what percentage of them would be mutated in breast cancer? This test needs to be run on the TCGA breast cancer data set with a couple of similar-sized lists of genes from a process unrelated to the DDR. If the results were that other similarly-sized sets give a much smaller %, it would greatly support the notion that the SLs uncovered with theta are enriched in DDR genes.

Typos and minor comments in the order they appear:

In. 37: The genes are known, it's their SL interaction that was not known.

In. 65: replace “;” with a “.” and change demonstrate to demonstrating in the next line (66).

In. 71-72: Rewrite sentence since yeast are organisms with no pol theta and do not show increased levels of spontaneous DNA damage. You mean Polq^{-/-} mutant cells: - change to: “Indeed, Polq^{-/-} cells demonstrate elevated levels of spontaneous DNA damage¹⁴.”

In. 75: In that study,...

In. 77: The end of this paragraph might be a good place to reference the recent work of Tijsterman/Kanaar (PMC5501794) and Saito, Maeda & Adachi (PMC5508229). It will also be necessary to discuss these two papers in the Discussion.

Ins. 80-83: Please add references for this statement.

In. 91: Reference for your mechanistic studies.

Ins. 195-197: To what are the authors referring for “...but deficient in 53BP1.” in this sentence?

In. 227: cacners -> cancers

In. 229: Change “TMEJ-mediated...” as the “M” in TMEJ means mediated

In. 232: Does Supplemental Table 3 have the data from which the 29.7% number was calculated? It is

not referred to in the sentence.

In. 298: suggests

In. 310: CO(subscript 2)

In. 312: lentiCRISPR

Ins. 701-710: Explain that the red triangles are the genes that you examine further in Fig. 1d.

Reviewer #2 (Remarks to the Author):

The manuscript by Feng et al. describes the results of a CRISPR-based synthetic lethality screen in search of genes that are functionally linked in some way to DNA polymerase theta expression and its role in DSB repair. The study design included appropriate control and reference samples, so as to verify the pol theta linkage to the cellular phenotype(s) used. As expected, a number of factors within the scope of the known components of the DNA repair machinery were found, and in addition a number of surprising genes were uncovered, including some in the DDR category and some in the metabolism and other categories. The results of this screen are interesting and will be useful to readers of the journal, and the manuscript, in general, is very well written and easy to follow.

The authors go on to describe experiments drilling down on and explaining one of the linkages uncovered, that between pol theta and the well-characterized DSB repair factor 53BP1. The results enabled them to propose a novel role for the 53BP1 interaction in the pol theta mediated DSB repair sub-pathway; this could be the target of future mechanistic studies. A possible therapeutic potential for the sub-pathway is discussed.

Reviewer #3 (Remarks to the Author):

Several labs have shown that cancer cells defective in homologous recombination are particularly sensitive to loss of DNA polymerase theta function, suggesting that pol theta could be a good chemotherapeutic target for these types of cancers. To identify other genes that might be essential for survival in the absence of pol theta, the authors utilized a CRISPR loss of function screen against selected DNA damage response (DDR) genes in MEFs. Quite surprisingly, they determined that 46% of all of the genes tested showed lethality/synthetic sickness, with genes from non-DSB repair GO categories fairly equally represented. They focused on 53BP1 as one representative PolqSL gene and, utilizing foci analysis, surmised that the lethality was due to an accumulation of aberrant HR intermediates. Importantly, they use computational analysis to show that TMEJ repair signatures were elevated in cancers with PolqSL mutations.

Overall, the findings are an exciting advance in the field and suggest that the importance of mammalian PolQ extends beyond two-ended double-strand break repair, into many contexts where impaired replication leads to fork collapse and subsequent breakage. I thought that the CRISPR screen was well-executed, although the significance of the various classes of PolqSL genes should be better described (see below). In addition, while the model is attractive, I'm not sure that the evidence for aberrant recombination intermediates is convincing and the authors might want to consider alternative explanations.

Major points

1. My biggest concern has to do with the authors' claim that the polq/53bp1 synthetic lethality (and other PolQSL lethality) is due to the accumulation of aberrant HR intermediates (Figs. 3 and 7). This is based

largely on the observation of an accumulation of Rad51 foci during S phase. However, the number of Rad51 foci per nucleus is typically less than one, even during late S phase. Do the authors postulate that each focus represents the coalescence of a number of DSBs? If not, it's hard to reconcile the small number of foci per nucleus with the severe synthetic lethality. Given that this is the only evidence for aberrant recombination intermediates, I wonder if the authors might be over-interpreting this or might have alternative models. It seems that there should be other damage tolerance mechanisms, such as translesion synthesis, that should be able to deal with base damage in the absence of BER/NER and PolQ.

2. Also, if the lethality is due to non-productive HR intermediates, then one might predict that structure-specific endonucleases (SSEs) might be required to deal with these intermediates. However, only *slx4* was identified as a PolqSL hit. Some clarification on this would be useful.

3. Although the novelty of the paper largely lies in the identification of PolqSL genes, the large number identified (46% of those tested) gives me pause. If non-DDR genes are included in the sgRNA pools (e.g. sugar metabolism, cytoskeleton, etc.), do any of those genes give positive synthetic lethal hits? Including these negative controls would increase confidence that the large number of identified DDR genes are biologically relevant and rule out (or perhaps rule in) the impact of loss of non-DNA metabolism genes in a *polq* mutant background.

4. Of the large number of PolqSL genes, the main focus is only on 53BP1 in the mechanistic analyses. Some further discussion of how other classes of genes fit into their model would be useful, especially for the chromatin structure and DDR signaling groups.

5. Figure 4: Another possible interpretation of the Figure 4 results is that PolQ is required for the bypass of mitomycin C-induced damage, rather than its repair. The increase in chromosome aberrations, Rad51/53BP1 foci, and SCEs could be due to increased DSBs that then must be repaired by recombination.

Minor points:

6. Some aspects of the model Figure 7 are confusing. In the first part of the figure, a DSB is created following replication fork collision. Is this an active, protein-mediated cleavage? Also, in the right branch that shows TMEJ, it isn't clear to me why MMEJ is shown operating on both DNA duplexes.

7. Figure 2: the time labels for the mitotic catastrophe and abnormal mitosis are confusing. The pictures always start at time=0, even though the cells are in different phases of the cell cycle. In the normal mitosis, the time period from late S to cytokinesis is about 5 hours, but it appears to take much longer in the abnormal mitosis. There is not point of reference in the middle panels (mitotic catastrophe) relative to S phase. Standardizing the time and phase labels for these panels would be helpful.

8. Figure 6B: What constitutes the 'other' class? This should be stated.

9. Shima et al. published no sensitivity of *polq* mutant mice or cells to MMC (PMID 15542845). The authors cite this reference and may want to comment on this discrepancy.

10. I couldn't find the exact nature of the *polq* null mutation anywhere in the paper or materials and methods. Although it is provided in prior studies, it should be included here as well.

11. Figure 1E: What is the common denominator for the 'DNA metabolism' class? Is this just a catch-all category?

12. Supplemental Table 2: In the first tab, it would be useful to have the genes grouped by the categories shown in the Euler diagram. Alternatively, the second tab could include the gene abundance change score for each gene. The significance of the color coding in the second tab isn't clear.

Response to Reviewers

We would like to thank the reviewers for their favorable response to our study, and for providing thoughtful and constructive critiques. We have considered each of their points in detail, and conducted several experiments to specifically address their points. We believe the revised manuscript is substantially improved and reveals several important insights regarding the biological role of Pol θ in mammalian DNA repair.

Perhaps most significantly, we conducted two additional CRISPR screens to provide further validation of the biological significance of the *Polq* synthetic lethal (*Polq*SL) genes that we identified in our study (Supplementary Figure 2). First, we demonstrate that there are no significant genes identified when the same DDR-CRISPR screen is performed in cells deficient for another DNA repair polymerase, Pol μ (*Polm*). Furthermore, we implemented a CRISPR screen targeting a distinct gene set (membrane proteins) in the *Polq*^{-/-} MEFs and found that only 2% of these genes had synthetic sickness/lethality, which is below the 3% false discovery rate cutoff used during statistical analyses. These findings solidify the main assertion of the paper: that while *Polq* is a non-essential gene, it becomes essential when a large number of DDR genes becomes mutated.

The other major improvement to our study pertains to the interpretation of the Rad51 foci that accumulate in *53bp1/Polq* double knockout cells. We previously interpreted this to suggest that *Polq* protects against “aberrant HR”. However, as Reviewer 3 correctly points out, there are alternative explanations that were incompletely considered in our initial submission. We have now challenged our *53bp1/Polq* double knockout MEFs with aphidicolin to induce replication fork stress/collapse (Figure 3e). We find that aphidicolin induces an even further increase in Rad51 foci, and that these foci are not resolved in the double knockout cells as they are in the *WT* or single knockout cells. These observations provide several important insights: 1) collapsed replication forks are a source of the rad51 foci observed in the *53bp1/Polq* DKO cells, and 2) rad51 foci represent a failed attempt at repair rather than an adaptive upregulation of HR in the DKO cells. Based on these findings, we suggest that there is a subset of replication-associated DSBs that is not amenable to HR repair, which must be repaired by Pol θ or 53BP1. We have revised our model in Figure 7 to reflect these insights.

Finally, with any exciting genetic screen study, there are many novel gene interactions that warrant mechanistic investigation. We believe that the most effective way to explore these is by sharing our screen results with the scientific community. In this revised study, we provide greater clarity on why we decided to investigate the synthetic lethality phenotype between *Polq* and *53bp1*. Although this interaction was previously reported in Wyatt et al, 2016, the mechanistic basis for this was not explored, and in fact, may have been incorrectly presumed to be related to NHEJ deficiency (which is not true, see Figure 2a-b). In the present study, all four of the *53bp1* pathway genes were highly significant and *53bp1* was universally amongst the top 10 gene interactions identified in each biological replicate that we performed. Because *53bp1* deficiency induces upregulation of both TMEJ and HR (Figure 2b), analysis of the *53bp1/Polq* DKO provides a unique opportunity to examine the non-redundant functions of TMEJ in a setting where HR is still active (and in fact hyperactive). This critically important (and novel) insight—that TMEJ provides an essential repair function that cannot be compensated by HR—also forms the basis for our assertion that hyperactive TMEJ can be selected for in cancers that do not have an underlying HR deficiency.

A point-by-point response to the reviewer critiques/comments is provided below.

Reviewer #1:

Q1: At the outset, it is not clear why so much effort is made to examine the 53BP1 SL, since this same group already reported this SL in an earlier study [Wyatt et al., (2016)].

A: Although we previously reported synthetic lethality between Polq and 53bp1 in a prior study, we had presumed that 53BP1 knockout phenocopied Ku70 deficiency due to an upstream deficiency in the NHEJ pathway. However, we demonstrate here (Figure 2b) that 53bp1 knockout cells remain NHEJ proficient at CRISPR-induced DSBs. By exploring the mechanism for 53bp1/Polq synthetic lethality in the present study, we demonstrate that a subset of replication-associated DSBs are irreparable by Rad51/HR and require Pol θ for effective repair. We believe that this biological function for Pol θ --to repair a subset of replication-associated DSBs that are not amenable for HR repair--explains the broad synthetic lethality with DDR factors that increase the levels of endogenous DNA damage, irrespective of the chemical/structural basis of that damage.

Q2: ...the old 53BP1 SL interaction is re-examined, then some experiments with MMC are introduced and next, yet another drug, PDS is examined for its effect in a Polq-/- background. All of these results are presented before the authors get to the cancer studies. In fact, before the cancer studies are presented, the authors shift to two more genes (Palb2 and Rad51), both of which again were also previously known to show SL with Polq. Overall, the fact that only a very few new PolqSL interactions are examined is a little disconcerting as it impacts the overall novelty of the work. As presented, the ms. appears to be cobbled together from several disparate pieces of work.

A: We have edited the text to provide a better transition between our 53bp1/polq synthetic lethal findings (Figures 2-3) and the MMC and PDS experiments (Figures 4-5). This is greatly facilitated by the addition of an aphidicolin experiment in Figure 3e, which demonstrates that *Polq* is required for repair of a subset of DSBs that are induced by replication fork collapse. We use MMC and PDS as different types of replication fork blocks that both illustrate an important role for Pol θ . The fact that Pol θ has an essential role in repair of replication-associated DSBs that is not compensated by HR is novel and a major finding of our study. We have validated several new PolqSL genes in Figure 1d, and in Supplementary Figure 3d-e. However, followup studies will be necessary to investigate individual gene interactions in greater mechanistic detail.

Q3: It certainly would have helped if a few new SL interactions were explored in greater detail.

A: During the course of performing experiments for the resubmission, we generated double knockout cells for 4 additional PolqSL genes and performed DNA repair foci studies. In all cases, interesting phenotypes were observed, yet each gene interaction required a series of followup experiments to gain the level of insights necessary for publication. We have included an example of increased endogenous 53bp1 foci after double knockout of *Neil3* and *Polq*

(Supplementary Figure 3d-e). Neil3 glycosylase has been implicated in DNA crosslink repair (PMID 27693351, Cell, 2016), and loss of Neil3 increases replication associated DSBs (PMID 29348879, Oncotarget, 2017). Our findings indicate that loss of both Neil3 and Polq results in higher levels of unrepaired DSBs. However, further investigation is necessary to understand the basis for this effect, and how it impacts repair of specific types of endogenous DNA damage.

Q4: On another note, in the cancer section on lns. 230-233: Is 30% actually a high number? If one were to take 309 genes from another non-essential process, what percentage of them would be mutated in breast cancer? This test needs to be run on the TCGA breast cancer data set with a couple of similar-sized lists of genes from a process unrelated to the DDR. If the results were that other similarly-sized sets give a much smaller %, it would greatly support the notion that the SLs uncovered with theta are enriched in DDR genes.

We were not claiming that the percentage of breast cancer DDR gene alterations is higher than anticipated by chance, since we do not have a suitable “control” cohort for this type of analysis. Our updated CRISPR screen using sgRNAs targeting genes encoding membrane proteins sheds light on this. Whereas 45% of the genes in the DDR-CRISPR library are synthetic lethal with *Polq*, only 2% of the genes in the membrane-CRISPR library are synthetic lethal with *Polq* (which is below the false discovery rate used in the statistical analyses). When we used 1000 randomly generated gene sets of equivalent size, we found that on average 23% of breast cancers had alteration of those genes, which is significantly lower than the 29.7% we observed with the PolqSL list. Furthermore, there was no association with microhomology-flanked deletions or COSMIC signature 3. Thus, we believe that the association between PolqSL gene alterations and TMEJ-associated genomic scars is specific for the DDR genes that we have identified. However, more specific biomarkers for hyper-active TMEJ in cancer are necessary, and will be the focus of a future investigation.

Typos and minor comments in the order they appear:

ln. 37: The genes are known, it's their SL interaction that was not known.

ln. 65: replace “;” with a “.” and change demonstrate to demonstrating in the next line (66).

ln. 71-72: Rewrite sentence since yeast are organisms with no pol theta and do not show increased levels of spontaneous DNA damage. You mean Polq-/- mutant cells: - change to: “Indeed, Polq-/- cells demonstrate elevated levels of spontaneous DNA damage¹⁴.”

ln. 75: In that study,...

ln. 80-83: Please add references for this statement.

ln. 91: Reference for your mechanistic studies.

ln. 195-197: To what are the authors referring for “...but deficient in 53BP1.” in this sentence?

ln. 227: cacners -> cancers

ln. 229: Change “TMEJ-mediated...” as the “M” in TMEJ means mediated

ln. 298: suggests

ln. 310: CO(subscript 2)

ln. 312: lentiCRISPR

ln. 701-710: Explain that the red triangles are the genes that you examine further in Fig. 1d.

Thank you for your suggestions, we have made all of these requested edits.

ln. 77: The end of this paragraph might be a good place to reference the recent work of Tijsterman/Kanaar (PMC5501794) and Saito, Maeda & Adachi (PMC5508229). It will also be necessary to discuss these two papers in the Discussion.

The ability of TMEJ to incorporate distant templates for repair using small tracts of microhomology, as in the case of random vector integration, is cited appropriately in the discussion. We did not find that placing it in the introduction was necessary.

ln. 232: Does Supplemental Table 3 have the data from which the 29.7% number was calculated? It is not referred to in the sentence.

Yes, the Supplementary Tables 4 and 5 (we have a new Supplementary Tables 3 for CRIPSR library) are now referred to in this sentence. These tables contain the necessary information to recreate the graphs shown in Figure 6f-j.

Reviewer #2:

Thank you very much for appreciating our work. There were no specific comments to address.

Reviewer #3:

Question 1 and 2: My biggest concern has to do with the authors' claim that the polq/53bp1 synthetic lethality (and other PolQSL lethality) is due to the accumulation of aberrant HR intermediates (Figs. 3 and 7). This is based largely on the observation of an accumulation of Rad51 foci during S phase. However, the number of Rad51 foci per nucleus is typically less than one, even during late S phase. Do the authors postulate that each focus represents the coalescence of a number of DSBs? If not, it's hard to reconcile the small number of foci per nucleus with the severe synthetic lethality. Given that this is the only evidence for aberrant recombination intermediates, I wonder if the authors might be over-interpreting this or might have alternative models. It seems that there should be other damage tolerance mechanisms, such as translesion synthesis, that should be able to deal with base damage in the absence of BER/NER and PolQ

A: We have conducted several experiments to address this point. First, we have quantified foci size to clarify that the thresholds that we are using for analysis of 53bp1/polq synthetic lethality measures Rad51 foci that are significantly larger than endogenous Rad51 foci that are observed during S phase (Supplementary Figure 6a-b). We cannot at this point distinguish whether these larger foci represent an aggregate of DSBs or increased Rad51 foci loading at one or two breaks.

However, by treating cells with aphidicolin and conducting a time course of release, we see a larger number of Rad51 foci that have delayed resolution in *53bp1/polq* double knockout cells (Figure 3e). This kinetic information indicates that the increases in Rad51 foci are not an adaptive response, but rather an accumulation due to ineffective repair. We thus refer to these findings as non-productive HR intermediates at replication-associated DSBs. We have substantially re-written our interpretation of these findings to better reflect the experimental data.

Q2. Also, if the lethality is due to non-productive HR intermediates, then one might predict that structure-specific endonucleases (SSEs) might be required to deal with these intermediates. However, only slx4 was identified as a PolqSL hit. Some clarification on this would be useful.

A: We do believe that the *slx4* interaction is related to this effect. Additionally, the increase in SCE's after MMC in *Polq*^{-/-} MEFs is also consistent with this phenomenon. The lack of finding other SSE nucleases in the PolqSL gene list is likely due to redundant activities. In unpublished collaborative work that is ongoing in *Drosophila* models, we find dramatic synthetic lethality between SSEs and *Polq* (*Mus308*), but only when multiple SSE's are disrupted.

Q3. Although the novelty of the paper largely lies in the identification of PolqSL genes, the large number identified (46% of those tested) gives me pause. If non-DDR genes are included in the sgRNA pools (e.g. sugar metabolism, cytoskeleton, etc.), do any of those genes give positive synthetic lethal hits? Including these negative controls would increase confidence that the large number of identified DDR genes are biologically relevant and rule out (or perhaps rule in) the impact of loss of non-DNA metabolism genes in a polq mutant background.

A: We employed two different approaches to address this question (see Supplementary Figure 2). First, we conducted a synthetic lethal screen using cells deficient for another DNA repair polymerase, Pol μ (*Polm*). Pol μ participates in NHEJ, but is not known to have a role in the repair of spontaneously arising replication-associated DSBs. Consistent with this, we identified no synthetic lethal interactions after performing the identical DDR-CRISPR process that in *Polq*^{-/-} MEFs revealed 140 synthetic lethal genes. For the second approach we obtained a CRISPR library from Addgene (ID: 1000000124) that targets 951 genes that encode membrane associated proteins. Performing the CRISPR synthetic lethal screen with this library in *WT*, *Polq*^{-/-}, and *Polq*^{hPOLQ} cells resulted in only 2% of the genes achieving statistical significance for synthetic lethal interaction. Because a 3% FDR was used in our analyses, it is likely that many of these represent false positive hits. Thus, the large percentage of synthetic lethal gene interactions identified in the DDR-CRISPR screen in *Polq*^{-/-} cells is reflective of *Polq* essentiality when a large proportion of DDR genes are mutated.

Q4. Of the large number of PolqSL genes, the main focus is only on 53BP1 in the mechanistic analyses. Some further discussion of how other classes of genes fit into their model would be useful, especially for the chromatin structure and DDR signaling groups.

A: We have re-written portions of the results and discussion to provide greater clarity regarding how we interpret this broad synthetic lethality with DDR genes (see pages 7-8 and 12-13). We believe that the revised manuscript more effectively highlights the link between *Polq* and repair

of replication-associated DSBs, in part due to the striking phenotype after aphidicolin treatment (see Supplementary Figure 3).

Q5. Figure 4: Another possible interpretation of the Figure 4 results is that PolQ is required for the bypass of mitomycin C-induced damage, rather than its repair. The increase in chromosome aberrations, Rad51/53BP1 foci, and SCEs could be due to increased DSBs that then must be repaired by recombination.

A: We do believe that *Polq* may facilitate bypass or tolerance of MMC damage, through a mechanism that is depicted in updated Figure 7. Collectively, our data argues for TMEJ repair of replication-associated DSBs, rather than a classical translesion bypass mechanism. However, further investigation using models that allow for visualization of ICLs would be necessary to truly distinguish between these possibilities.

Minor points:

Q6. Some aspects of the model in Figure 7 are confusing. In the first part of the figure, a DSB is created following replication fork collision. Is this an active, protein-mediated cleavage? Also, in the right branch that shows TMEJ, it isn't clear to me why MMEJ is shown operating on both DNA duplexes.

A: Thank you for this comment. We have revised our model to incorporate recent evidence for inheritance of replication blocking lesions that fits nicely into our functional data and similar findings that have been made for TMEJ/Pol θ functions in *C. Elegans*. In the prior schematic, Pol θ was only acting on one of the duplexes, and we have improved the figure to avoid this confusion.

Q7. Figure 2: the time labels for the mitotic catastrophe and abnormal mitosis are confusing. The pictures always start at time=0, even though the cells are in different phases of the cell cycle. In the normal mitosis, the time period from late S to cytokinesis is about 5 hours, but it appears to take much longer in the abnormal mitosis. There is not point of reference in the middle panels (mitotic catastrophe) relative to S phase. Standardizing the time and phase labels for these panels would be helpful.

A: Thank you for noting this. We have made the requested change.

Q8. Figure 6B: What constitutes the 'other' class? This should be stated.

A: The "other" category is all non-TMEJ deletions that are larger than >5bp. This is now explicitly defined in the Figure legend.

Q9. Shima et al. published no sensitivity of polq mutant mice or cells to MMC (PMID 15542845). The authors cite this reference and may want to comment on this discrepancy.

A: We have cited this study and included a discussion of this discrepancy (revised page 9). We believe the difference is related to residual Pol θ activity from their hypomorphic *Polq* allele. There is even mention in their discussion that they see MMC sensitivity of the true Polq KO ES cells, although that data is not included in the paper.

Q10. I couldn't find the exact nature of the polq null mutation anywhere in the paper or materials and methods. Although it is provided in prior studies, it should be included here as well.

A: We have included a citation for the origin of the *Polq*^{-/-} MEFs in the methods section.

Q11. Figure 1E: What is the common denominator for the 'DNA metabolism' class? Is this just a catch-all category?

A: This is a catch-all category of genes encoding proteins that directly modify DNA but are not specific to a particular DNA repair pathway. Included here are helicases, nucleases, and methyltransferases that directly metabolize or modify DNA.

Q12. Supplemental Table 2: In the first tab, it would be useful to have the genes grouped by the categories shown in the Euler diagram. Alternatively, the second tab could include the gene abundance change score for each gene. The significance of the color coding in the second tab isn't clear.

A: We have updated Supplementary Table 2 to make it easier for the reader to identify genes and their CRISPR Abundance change scores. Thank you for noting this issue. We have also removed any unnecessary color coding.

REVIEWERS' COMMENTS:

Reviewer #3 (Remarks to the Author):

The authors have done an admirable job responding to the reviewer critiques. I believe that the manuscript is significantly improved and all my questions have been addressed. The addition of the pol μ and membrane protein CRISPR screens greatly increases my confidence in the authors' interpretations and the rewriting has made the story more logical. Two small points remain:

1. In the description of the Neil3 validation (lines 163-165), it isn't immediately clear why loss of a DNA glycosylase should result in an accumulation of unrepaired replication associated DSBs. The authors' response to reviewer question 3 makes this clearer and they may want to consider including an abbreviated explanation and/or the Oncotarget reference (PMID 29348879) in this section.
2. Supplemental Figure 6 legend: change "...which eliminates them majority..." to "the majority".

Responses to the reviewers:
Reviewer #3

The authors have done an admirable job responding to the reviewer critiques. I believe that the manuscript is significantly improved and all my questions have been addressed. The addition of the polμ and membrane protein CRISPR screens greatly increases my confidence in the authors' interpretations and the rewriting has made the story more logical.

A: Thank you very much for appreciating our work.

Two small points remain:

1. In the description of the Neil3 validation (lines 163-165), it isn't immediately clear why loss of a DNA glycosylase should result in an accumulation of unrepaired replication associated DSBs. The authors' response to reviewer question 3 makes this clearer and they may want to consider including an abbreviated explanation and/or the Oncotarget reference (PMID 29348879) in this section.

A: Thank you for your suggestion, we have a reference to the Oncotarget paper in the text.

2. Supplemental Figure 6 legend: change "...which eliminates them majority..." to "the majority".

A: Thank you for your suggestion, we have made the correction.